# Bootstrap Learning for Combinatorial Graph Alignment with Sequential GNNs

## Abstract

Graph neural networks (GNNs) have struggled to outperform traditional optimization methods on combinatorial problems, limiting their practical impact. We address this gap by introducing a novel chaining procedure for the graph alignment problem—a fundamental NP-hard task of finding optimal node correspondences between unlabeled graphs using only structural information.

Our method trains a sequence of GNNs where each network learns to iteratively refine similarity matrices produced by previous networks. During inference, this creates a bootstrap effect: each GNN improves upon partial solutions by incorporating discrete ranking information about node alignment quality from prior iterations. We combine this with a powerful architecture that operates on node pairs rather than individual nodes, capturing global structural patterns essential for alignment that standard message-passing networks cannot represent.

Extensive experiments on synthetic benchmarks demonstrate substantial improvements: our chained GNNs achieve over 3× better accuracy than existing methods on challenging instances, and uniquely solve regular graphs where all competing approaches fail. When combined with traditional optimization as post-processing, our method substantially outperforms state-of-the-art solvers on the graph alignment benchmark.

## 1 Introduction

"Combinatorial optimization searches for an optimum object in a finite collection of objects. Typically, the collection has a concise representation (like a graph), while the number of objects is huge."(Schrijver et al., 2003) This field bridges discrete mathematics, mathematical programming, and computer science, with applications spanning logistics, network design, and resource allocation. Machine learning offers a promising approach to combinatorial optimization (CO) by exploiting patterns in problem instances to design faster algorithms for specific problem families (Bengio et al., 2021). Graph neural networks (GNNs) emerge as natural tools for this integration, given the inherently discrete and graph-structured nature of most CO problems (Cappart et al., 2023).

**Limited success of learning approaches.** Despite significant research efforts, GNN-based methods have struggled to outperform traditional specialized solvers on most CO problems. The traveling salesperson problem exemplifies this challenge—while receiving substantial attention since (Vinyals et al., 2015), GNN approaches remain limited to small-scale instances. Similarly, simple greedy heuristics continue to outperform sophisticated GNNs on problems like maximum independent set (Angelini & Ricci-Tersenghi, 2023; Böther et al., 2022).

The graph matching problem can be cast as a combinatorial graph alignment problem (GAP). Machine learning methods have been widely applied in related areas such as pattern recognition (Conte et al., 2004), computer vision (Sun et al., 2020), and social network analysis (Narayanan & Shmatikov, 2008) (see Section A.6 for further discussion). Their motivation is that in noisy real-world data the ground-truth matching may deviate from the mathematically optimal solution, making it more effective to learn a matching directly from data. In this work, however, we focus strictly on the combinatorial optimization setting, where only the mathematically optimal solution is relevant. Accordingly, we use the term *graph alignment* rather than *graph matching*.

The graph alignment problem (GAP) provides an ideal testbed for exploring GNN capabilities in CO. GAP seeks the node correspondence between two graphs that maximally aligns their edge

Table 1: Approximation quality $\frac{\text{ALG}}{\text{OPT}}$ for sparse, dense and regular random graphs. **Proj** and **FAQ** are used to produce a permutation from the convex relaxation solution $D_{\text{cx}}$ or from the similarity matrix computed by FGNN or chained FGNN (ChFGNN).

APPROXIMATION QUALITY $\frac{\text{ALG}}{\text{OPT}}$ FOR RANDOM GRAPHS (IN %).

| TYPE OF GRAPHS | | SPARSE | DENSE | REGULAR |
|---|---|---|---|---|
| BASELINES | **PROJ**($D_{\text{cx}}$) | 17.3 | 24.4 | 2.9 |
| (NON-NEURAL) | **FAQ**($D_{\text{cx}}$) | 67.1 | 53 | 27 |
| BASELINES | FGNN **PROJ** | 17.8 | 23.6 | 6.7 |
| (NEURAL) | FGNN **FAQ** | 71.1 | 47 | 54 |
| CHAINING | CHFGNN **PROJ** | 95.8 | 44 | 67.1 |
| CHAINING | CHFGNN **FAQ** | **98.8** | **77.4** | **81.8** |

structures—a fundamental problem encompassing graph isomorphism as a special case. In its general form, GAP reduces to the NP-hard quadratic assignment problem (QAP).

**Iterative refinement through chaining.** We introduce a novel technique—*chaining of GNNs*—that for the first time demonstrates GNN methods outperforming state-of-the-art specialized solvers on the combinatorial graph alignment problem. Our approach combines multiple GNNs in an iterative refinement procedure, with each network learning to improve upon the previous iteration's solution. Our chaining procedure trains a sequence of GNNs where each network learns to enhance partial solutions produced by previous networks. This creates a bootstrap effect during inference, where GNNs iteratively refine alignment estimates. The approach can be combined with traditional solvers like the Frank-Wolfe-based **FAQ** algorithm (Vogelstein et al., 2015), creating hybrid methods that outperform both pure learning and pure optimization approaches.

Table 1 illustrates our key results across different graph types. To evaluate how close our algorithm is to the best possible solution, we measure its *approximation quality*[1] as $\frac{\text{ALG}}{\text{OPT}}$ in percent, where ALG is the number of aligned edges obtained by our algorithm and OPT is the number of aligned edges of the optimal solution. A score of $100\%$ corresponds to optimality, and lower values indicate a smaller fraction of the optimal alignment achieved. Our chained GNNs, particularly when coupled with **FAQ** post-processing (ChGNN **FAQ**), consistently achieve the best performance.

We use synthetic datasets for both training and evaluation to control problem difficulty and assess generalization. In doing so, we follow the standard benchmarking methodology of combinatorial optimization, which favors randomly generated instances (Skorin-Kapov, 1990; Taillard, 1991). Unlike real-world data, which is often either too trivial or intractably difficult, synthetic instances enable more robust and fine-grained comparisons between algorithms. Finally, we confirm the effectiveness and transferability of our method by achieving strong results on three real-world graph pairs (biology, social networks, and road networks), thereby validating our findings from synthetic data.

We address the graph alignment problem, which we formulate as a machine learning task in Section 2. While traditional optimization methods have so far surpassed learning-based approaches for this problem (Section 4), we introduce a method that reverses this trend. Our main contribution is a novel training and inference procedure, the chaining procedure, where sequential GNNs learn to improve partial solutions through iterative refinement (Section 3). This procedure leverages a modified Folklore-type GNN architecture with enhanced expressiveness (Section A.4), making it particularly effective on challenging regular graphs where standard methods fail. As demonstrated in Section 5, our chained GNNs coupled with **FAQ** post-processing, outperform all existing solvers on synthetic graph alignment benchmarks. These findings suggest that iterative refinement via chained learning offers a promising general framework for advancing GNN performance on other combinatorial optimization (CO) problems, potentially bridging the gap between machine learning and traditional optimization.

**Mathematical notations.** Let $G = (V, E)$ be a simple graph with $V = \{1, \ldots, n\}$ and adjacency matrix $A \in \{0, 1\}^{n \times n}$, where $A_{ij} = 1$ if $(i, j) \in E$ and 0 otherwise. Let $\mathcal{S}_n$ denote the set of permutations of $V$, with each $\pi \in \mathcal{S}_n$ associated to a permutation matrix $P \in \{0, 1\}^{n \times n}$ defined by

---

[1]In the algorithms literature, the approximation ratio is traditionally written as $\frac{\text{OPT}}{\text{ALG}} \geq 1$, so that an algorithm is called a $k$-approximation if $\text{OPT}/\text{ALG} \leq k$. We instead adopt the $\frac{\text{ALG}}{\text{OPT}}$ formulation, which is more in line with evaluation metrics in machine learning, where higher scores denote better performance.

$P_{ij} = 1$ iff $\pi(i) = j$. The set of doubly stochastic matrices is denoted $\mathcal{D}_n$. For $A, B \in \mathbb{R}^{n \times n}$, the Frobenius inner product and norm are $\langle A, B \rangle = \text{trace}(A^\top B)$ and $\|A\|_F = \sqrt{\langle A, A \rangle}$, respectively.

## 2 FROM COMBINATORIAL OPTIMIZATION TO LEARNING

This section introduces the graph alignment problem (GAP) from a combinatorial optimization perspective, presents the state-of-the-art **FAQ** algorithm, and describes how we formulate GAP as a learning problem using synthetic datasets with controllable difficulty.

### 2.1 GRAPH ALIGNMENT IN COMBINATORIAL OPTIMIZATION

**Problem formulation.** Given two $n \times n$ adjacency matrices $A$ and $B$ representing graphs $G_A$ and $G_B$, the graph alignment problem seeks to find the permutation that best aligns their structures. Formally, we minimize the Frobenius norm:

$$\text{GAP}(A, B) = \min_{\pi \in \mathcal{S}_n} \sum_{i,j} \left(A_{ij} - B_{\pi(i)\pi(j)}\right)^2 = \min_{P \in \mathcal{S}_n} \|AP - PB\|_F^2, \tag{1}$$

where we used the identity $\|A - PBP^T\|_F^2 = \|AP - PB\|_F^2$ for permutation matrices $P$. Expanding the right-hand term, we see that minimizing (1) is equivalent to maximizing the number of matched edges:

$$\max_{P \in \mathcal{S}_n} \langle AP, PB \rangle = \max_{\pi \in \mathcal{S}_n} \sum_{i,j} A_{ij} B_{\pi(i)\pi(j)}. \tag{2}$$

This formulation connects GAP to the broader class of Quadratic Assignment Problems (QAP) (Burkard et al., 1998).

**Computational complexity.** The GAP is computationally challenging, as it reduces to several well-known NP-hard problems. For instance, when $G_A$ has $n$ vertices and $G_B$ is a single path or cycle, GAP becomes the Hamiltonian path/cycle problem. When $G_B$ consists of two cliques of size $n/2$, we recover the minimum bisection problem. More generally, solving (1) is equivalent to finding a maximum common subgraph, which is APX-hard (Crescenzi et al., 1995).

**Performance metrics.** We denote an optimal solution as $\pi^{A \to B}$. We evaluate alignment quality using two complementary metrics (that should be maximized):

$$\text{Accuracy:} \quad \mathbf{acc}(\pi, \pi^{A \to B}) = \frac{1}{n} \sum_{i=1}^{n} \mathbf{1}(\pi(i) = \pi^{A \to B}(i)) \tag{3}$$

$$\text{Number of common edges:} \quad \mathbf{nce}(\pi) = \frac{1}{2} \sum_{i,j} A_{ij} B_{\pi(i)\pi(j)} \tag{4}$$

Accuracy measures the fraction of correctly matched nodes, while the number of common edges quantifies structural similarity. In Table 1, the ratio $\frac{\text{ALG}}{\text{OPT}}$ is computed as $\frac{\mathbf{nce}(\pi^{\text{ALG}})}{\mathbf{nce}(\pi^{A \to B})}$. Note that even if this ratio is one, the accuracy may still be low if the GAP has no unique solution (as illustrated on real datasets in Section 5.5).

### 2.2 CONTINUOUS RELAXATIONS AND THE **FAQ** ALGORITHM

**Relaxation approach.** Since the discrete optimization in (1) is intractable, we consider continuous relaxations where the discrete permutation set $\mathcal{S}_n$ is replaced by the continuous set of doubly stochastic matrices $\mathcal{D}_n$ in (1) or (2):

- **Convex relaxation**:

$$\arg \min_{D \in \mathcal{D}_n} \|AD - DB\|_F^2 = D_{\text{cx}} \tag{5}$$

  This yields a convex optimization problem with guaranteed global optimum.

- **Indefinite relaxation**:

$$\max_{D \in \mathcal{D}_n} \langle AD, DB \rangle \tag{6}$$

  This non-convex formulation often provides better solutions but is NP-hard in general due to its indefinite Hessian (Pardalos & Vavasis, 1991).

**Solution extraction.** Both relaxations produce doubly stochastic matrices $D$ that must be projected to permutation matrices. This projection solves the linear assignment problem $\max_{P \in \mathcal{S}_n} \langle P, D \rangle$, efficiently solved by the Hungarian algorithm in $O(n^3)$ time (Kuhn, 1955). We denote this projection as **Proj**$(D) \in \mathcal{S}_n$.

**FAQ algorithm.** The Fast Approximate Quadratic (**FAQ**) algorithm proposed by Vogelstein et al. (2015) approximately solves the indefinite relaxation (6) using Frank-Wolfe optimization and then projects this solution in $\mathcal{S}_n$. Unlike the convex relaxation, **FAQ**'s performance depends critically on initialization. We denote the **FAQ** solution with initial condition $D$ as **FAQ**$(D) \in \mathcal{S}_n$. As demonstrated in Lyzinski et al. (2015), **FAQ** often significantly outperforms simple projection: **FAQ**$(D_{\mathrm{cx}})$ typically yields much better solutions than **Proj**$(D_{\mathrm{cx}})$, especially for challenging instances. **This improvement motivates our approach of providing FAQ with better initializations through learned similarity matrices.**

### 2.3 SYNTHETIC DATASETS: CONTROLLED DIFFICULTY THROUGH NOISE

**Connection to graph isomorphism.** When graphs $G_A$ and $G_B$ are isomorphic (GAP$(A, B) = 0$), the alignment problem reduces to graph isomorphism (GI). While GI's complexity remains open—it's neither known to be in P nor proven NP-complete—Babai (2016)'s recent breakthrough shows it's solvable in quasipolynomial time. We study a natural generalization: noisy graph isomorphism, where noise level controls problem difficulty. At zero noise, graphs are isomorphic; as noise increases, they become increasingly different, making alignment more challenging.

**Correlated random graph model.** Our datasets consist of correlated random graph pairs $(G_A, G_B)$ with identical marginal distributions but controllable correlation. This design allows systematic difficulty variation while maintaining statistical properties. The generation process involves: (i) Create correlated graphs $G_A$ and $G_B$ with known alignment; (ii) Apply random permutation $\pi^\star \in \mathcal{S}_n$ to $G_B$, yielding $G_B'$: (iii) Use triplets $(G_A, G_B', \pi^\star)$ for supervised learning.

We employ three graph families—**Bernoulli**, **Erdős-Rényi**, and **Regular**—with parameters: **Number of nodes**: $n$; **Average degree**: $d$; **Noise level**: $p_{\mathrm{noise}} \in [0, 1]$, see Section A.1 for precise definitions. The noise parameter controls edge correlation: the graphs $G_A$ and $G_B$ (before applying the random permutation) share $(1 - p_{\mathrm{noise}})nd/2$ edges on average (with $p_{\mathrm{noise}} = 0$ yielding isomorphic graphs). For low noise levels, we expect $\pi^\star = \pi^{A \to B}$, providing clean supervision. However, for high noise, the planted permutation $\pi^\star$ may not be optimal, introducing label noise that makes learning more challenging.

## 3 LEARNING THROUGH CHAINING

**Overview.** The chaining procedure works by iteratively refining graph alignment estimates through three key operations: (1) computing node similarities, (2) extracting and evaluating the current best permutation, and (3) using this evaluation to generate improved node features. Each iteration produces a better similarity matrix, leading to more accurate alignments.

### 3.1 CHAINING PROCEDURE

**Step 1: Initial feature extraction and similarity computation.** Given a mapping $f$ that extracts node features from a graph's adjacency matrix $A \in \{0, 1\}^{n \times n}$ and outputs $f : \{0, 1\}^{n \times n} \to \mathbb{R}^{n \times d}$, we compute node feature matrices $f(A)$ and $f(B)$ for graphs $G_A$ and $G_B$. The initial similarity matrix captures pairwise node similarities via their feature dot products:

$$S^{A \to B, (0)} = f(A)f(B)^T \in \mathbb{R}^{n \times n}. \tag{7}$$

Here, $S_{ij}^{A \to B, (0)}$ measures the similarity between node $i \in G_A$ and node $j \in G_B$ based on their learned features.

**Step 2: Permutation extraction and node quality scoring.** From a similarity matrix $S^{A \to B}$, we extract the best permutation estimate by solving the linear assignment problem: $\pi = \textbf{Proj}(S^{A \to B})$ where $\pi = \arg\max_{\pi \in \mathcal{S}_n} \sum_i S_{i\pi(i)}^{A \to B}$. This permutation $\pi : G_A \to G_B$ represents our current best guess for the optimal alignment $\pi^{A \to B}$. To evaluate alignment quality, we compute a score for each node $i$ in graph A: $\mathrm{score}(i) = \sum_j A_{ij} B_{\pi(i)\pi(j)}$. Intuitively, $\mathrm{score}(i)$ counts the number

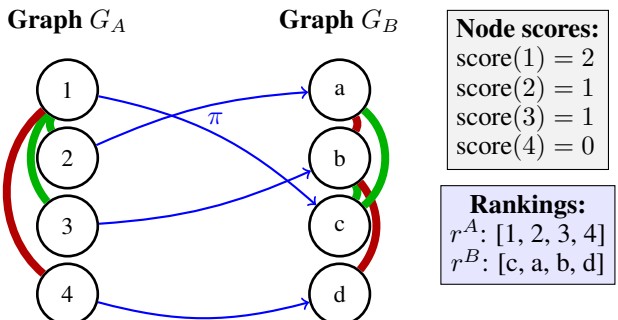

Figure 1: Illustration of Step 2. The permutation $\pi$ maps 1→c, 2→a, 3→b, 4→d. Green edges show matches: edge 1-2 with a-c, and edge 1-3 with b-c. Node 1 has the highest score (2 matched edges), nodes 2 and 3 each have 1 matched edge, and node 4 has no matched edges.

of edges incident to node $i$ that are correctly matched under the current permutation $\pi$—higher scores indicate better-aligned nodes (see Figure 1). We then rank nodes in $G_A$ by decreasing score, obtaining a ranking $r^A \in \mathcal{S}_n$ such that:

$$\text{score}(r^A(1)) \geq \text{score}(r^A(2)) \geq \cdots \geq \text{score}(r^A(n)). \tag{8}$$

The corresponding ranking for $G_B$ is derived as $r^B(i) = \pi(r^A(i))$, ensuring that highly-ranked nodes in both graphs correspond to each other under the current permutation (see Figure 1). Note that when inequalities in (8) are strict, the rankings uniquely encode the permutation $\pi$ (with top-ranked nodes being those most reliably aligned).

**Step 3: Ranking-enhanced feature learning.** We now incorporate the ranking information to compute improved node features. Using a mapping $g : \{0,1\}^{n \times n} \times \mathcal{S}_n \to \mathbb{R}^{n \times d}$ that takes both the graph structure and node rankings as input, we compute enhanced feature matrices $g(A, r^A)$ and $g(B, r^B)$. The new similarity matrix is:

$$S^{A \to B} = g(A, r^A) g(B, r^B)^T \in \mathbb{R}^{n \times n}. \tag{9}$$

This ranking-enhanced similarity matrix $S^{A \to B,(1)} = g(A, r^{A,(0)}) g(B, r^{B,(0)})^T$ should be more informative than the initial $S^{A \to B,(0)}$ since it incorporates knowledge about which nodes align well. Consequently, we expect **Proj**$(S^{A \to B,(1)})$ to be closer to the optimal $\pi^{A \to B}$ than **Proj**$(S^{A \to B,(0)})$.

**Iterative refinement.** The key insight is to iterate steps 2 and 3 (see Figure 2) with different learned mappings $g^{(1)}, g^{(2)}, \ldots$ at each iteration, progressively improving the similarity matrix and resulting permutation. This creates a bootstrap effect where each iteration leverages the improved alignment from the previous step. The complete chaining procedure requires a sequence of mappings:

$$f : \{0,1\}^{n \times n} \to \mathbb{R}^{n \times d}, r : \{0,1\}^{n \times n} \times \{0,1\}^{n \times n} \times \mathbb{R}^{n \times n} \to \mathcal{S}_n \times \mathcal{S}_n, \tag{10}$$

$$g^{(1)} : \{0,1\}^{n \times n} \times \mathcal{S}_n \to \mathbb{R}^{n \times d}, g^{(2)} : \{0,1\}^{n \times n} \times \mathcal{S}_n \to \mathbb{R}^{n \times d}, \quad \ldots \tag{11}$$

The procedure flows as follows: $f$ computes the initial similarity matrix $S^{A \to B,(0)}$ via (7), then $r$ computes rankings $r^{A,(0)}, r^{B,(0)}$ via (8), then $g^{(1)}$ computes the refined similarity matrix $S^{A \to B,(1)}$ via (9), and so forth, see Figure 2.

## 3.2 TRAINING AND INFERENCE WITH CHAINED GNNS

The ranking step $r$ is not differentiable, preventing end-to-end training. Instead, we train each GNN in the chain sequentially, which proves both practical and effective. This approach allows our method to explicitly learn from discrete permutation decisions at each step, which is crucial for the iterative improvement process.

**Sequential training procedure.** The mappings $f, g^{(1)}, g^{(2)}, \ldots$ are implemented using graph neural networks (GNNs). We train the GNNs $f, g^{(1)}, g^{(2)}, \ldots, g^{(k)}$ sequentially, where each network is optimized to improve upon the previous iteration's output. For training data consisting of graph pairs $(G_A, G_B)$ with known ground truth permutation $\pi^\star$, we define a cross-entropy loss for any similarity matrix $S^{A \to B}$: $\mathcal{L}(S^{A \to B}, \pi^\star) = -\sum_i \log \left(\text{softmax}(S^{A \to B})\right)_{i\pi^\star(i)}$. This loss encourages

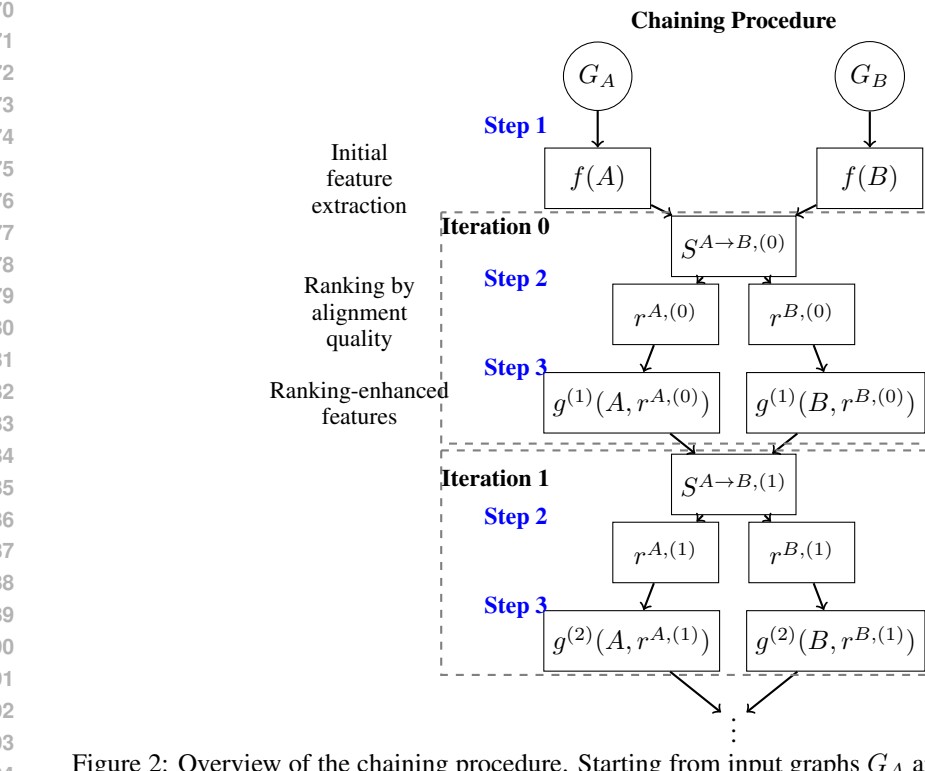

Figure 2: Overview of the chaining procedure. Starting from input graphs $G_A$ and $G_B$, we first (1) extract features and compute similarities, then iteratively (2) rank nodes by alignment quality, and (3) use rankings to enhance features and similarities.

the similarity matrix to assign high values to the correct node correspondences specified by $\pi^\star$. The training proceeds as follows:

1. **Train $f$**: Minimize $\mathcal{L}(S^{A \to B,(0)}, \pi^\star)$ to learn initial feature extraction.

2. **Train $g^{(1)}$**: Fix $f$, compute $r^{A,(0)}$ and $r^{B,(0)}$ for the training data, then minimize $\mathcal{L}(S^{A \to B,(1)}, \pi^\star)$.

3. **Train $g^{(2)}$**: Fix $f$, compute $r^{A,(0)}, r^{B,(0)}$ then fix $g^{(1)}$, compute $r^{A,(1)}, r^{B,(1)}$, then minimize $\mathcal{L}(S^{A \to B,(2)}, \pi^\star)$.

4. **Continue**: Repeat this process for $g^{(3)}, g^{(4)}, \dots, g^{(k)}$.

This sequential approach ensures that each GNN learns to improve upon the alignment quality achieved by all previous networks in the chain.

**Inference procedure.** During inference on new graph pairs $(G_A, G_B)$, we apply the trained networks sequentially: $f$ produces $S^{A \to B,(0)}$, then alternating applications of $r$ and $g^{(\ell)}$ produce refined similarity matrices $S^{A \to B,(1)}, S^{A \to B,(2)}, \dots, S^{A \to B,(L)}$. Each similarity matrix $S^{A \to B,(\ell)}$ represents a progressively better estimate of node correspondences. To extract a discrete permutation from any $S^{A \to B,(\ell)}$, we apply either the Hungarian algorithm **Proj** or the **FAQ** algorithm, yielding candidate permutation $\pi^{(\ell)}$. We can then estimate its performance by computing $\mathbf{nce}(\pi^{(\ell)})$ defined in (4).In practice, we observe that $\mathbf{nce}(\pi^{(\ell)})$ typically increases with $\ell$, confirming that each iteration improves alignment quality.

**Looping for enhanced performance.** An important observation is that the final trained network $g^{(L)}$ can be applied multiple times to further refine the solution. Since $g^{(L)}$ is trained to improve partial solutions, repeatedly applying $g^{(L)}$ (with intermediate ranking steps $r$) often yields additional improvements. We call this technique **looping** and explore its benefits in Section 5.2. This allows us to achieve better performance without training additional networks, simply by iterating the refinement process as long as the number of common edges increases.

### 3.3 GNN ARCHITECTURE AND EXPRESSIVENESS

**Architecture choice and motivation.** We implement all GNN mappings $f$, $g^{(1)}$, $g^{(2)}$, ..., $g^{(k)}$ using the same architecture inspired by Folklore-type GNNs (Maron et al., 2019). Unlike standard message passing neural networks (MPNNs), this architecture operates on node pairs rather than individual nodes, providing greater expressiveness at the cost of scalability (Maron et al., 2019).

**Core architecture: Folklore-inspired residual layers.** Our GNN's main building block is a residual layer that processes hidden states for all node pairs $(h_{i \to j}^t)_{i,j} \in \mathbb{R}^{n \times n \times d}$, producing updated states $(h_{i \to j}^{t+1})_{i,j} \in \mathbb{R}^{n \times n \times d}$: $h_{i \to j}^{t+1} = h_{i \to j}^t + m_1 \left( h_{i \to j}^t, \sum_\ell h_{i \to \ell}^t \odot m_0(h_{\ell \to j}^t) \right)$, where $m_0 : \mathbb{R}^d \to \mathbb{R}^d$ and $m_1 : \mathbb{R}^{2d} \to \mathbb{R}^d$ are multilayer perceptrons (MLPs) with graph normalization layers, and $\odot$ denotes component-wise multiplication. We refer to Section A.4 for more details about our FGNN.

## 4 RELATED WORK: STATE-OF-THE-ART AND LEARNING LIMITATIONS

Additional related work on machine learning approaches to graph matching is discussed in Section A.6. In this section, we restrict our attention to the combinatorial optimization perspective.

**Non-learning methods.** Among traditional optimization approaches, **FAQ** represents the state-of-the-art for correlated random graphs (Lyzinski et al., 2015), outperforming the convex relaxation, GLAG algorithm (Fiori et al., 2013), PATH algorithm (Zaslavskiy et al., 2008), Umeyama's spectral method (Umeyama, 1988), and linear programming approaches (Almohamad & Duffuaa, 1993). More recent papers (Xu et al., 2019) and (Bommakanti et al., 2024) proposed new algorithms for GAP but their comparison with **FAQ** is not correct probably because they used a suboptimal initialization (see more details in Section A.3)

**Learning approaches and their limitations.** Recent GNN-based methods for graph alignment include approaches by Yu et al. (2023), PGM (Kazemi et al., 2015), MGCN (Chen et al., 2020), and MGNN (Wang et al., 2021). For Erdős-Rényi graphs, none of these methods demonstrated positive accuracy under the same noise level where our experimental results show **FAQ**($D_{\text{cx}}$) maintained positive accuracy (see Section A.1).

This analysis reveals a significant gap: **before our work, FAQ($D_{\text{cx}}$) represented the state-of-the-art for GAP on correlated random graphs, substantially outperforming all existing learning and GNN approaches.** Our chaining procedure aims to bridge this gap by combining the expressiveness of GNNs with iterative refinement, ultimately providing **FAQ** with superior initializations that improve upon both pure learning and pure optimization approaches.

## 5 EMPIRICAL RESULTS AND COMPARISON TO **FAQ**

We evaluate our chaining procedure against **FAQ** Vogelstein et al. (2015), which represents the state-of-the-art for graph alignment on correlated random graphs. We implement all GNN mappings $f$, $g^{(1)}$, $g^{(2)}$, ..., $g^{(k)}$ using the same architecture inspired by Folklore-type GNNs (Maron et al., 2019). Our experiments compare three categories of methods: (1) non-neural baselines using convex relaxation, (2) neural baselines using single-step FGNNs, and (3) our chained FGNNs with iterative refinement. All methods can be combined with **Proj** and **FAQ** as a post-processing step to extract a permutation (see Section 2.2).

### 5.1 MAIN RESULTS ON SYNTHETIC DATASETS

Table 2 presents comprehensive results across different graph types (with 500 nodes) and noise levels. We evaluate on three challenging scenarios: sparse Erdős-Rényi graphs (average degree 4), dense Erdős-Rényi graphs (average degree 80), and regular graphs (degree 10). The noise parameter $p_{\text{noise}}$ controls the difficulty, with higher values indicating more corrupted alignments.

**Sparse and dense Erdős-Rényi graphs.** For both sparse and dense graphs, our chained FGNNs significantly outperform all baselines, particularly at challenging noise levels. At $p_{\text{noise}} = 0.25$, chained FGNNs with **FAQ** post-processing achieve 85% accuracy on sparse graphs, compared to

Table 2: Accuracy (**acc**) defined in (3) for Erdős-Rényi and regular graphs as a function of the noise $p_{\text{noise}}$. FGNN refers to the architecture in Section A.4 and ChFGNN to our chained FGNNs. **Proj** and **FAQ** are used to produce a permutation (from the similarity matrix computed).

| SPARSE ERDŐS-RÉNYI GRAPHS WITH AVERAGE DEGREE 4 | | | | | | | | | |
|---|---|---|---|---|---|---|---|---|---|
| ER 4 (**ACC**) | NOISE | 0 | 0.05 | 0.1 | 0.15 | 0.2 | 0.25 | 0.3 | 0.35 |
| BASELINES | **PROJ**($D_{\text{cx}}$) | **0.98** | **0.97** | 0.90 | 0.59 | 0.23 | 0.09 | 0.04 | 0.02 |
| (NON-NEURAL) | **FAQ**($D_{\text{cx}}$) | **0.98** | **0.98** | **0.96** | **0.95** | 0.73 | 0.13 | 0.04 | 0.02 |
| BASELINES | FGNN **PROJ** | 0.98 | 0.94 | 0.74 | 0.44 | 0.23 | 0.12 | 0.06 | 0.03 |
| (NEURAL) | FGNN **FAQ** | 0.98 | 0.98 | 0.96 | 0.95 | 0.81 | 0.24 | 0.07 | 0.03 |
| CHAINING | CHFGNN **PROJ** | 0.98 | 0.98 | 0.96 | 0.94 | 0.91 | 0.82 | 0.49 | 0.08 |
| CHAINING | CHFGNN **FAQ** | 0.98 | 0.98 | 0.96 | 0.95 | **0.93** | **0.85** | **0.52** | 0.09 |

| DENSE ERDŐS-RÉNYI GRAPHS WITH AVERAGE DEGREE 80 | | | | | | | | | |
|---|---|---|---|---|---|---|---|---|---|
| ER 80 (**ACC**) | NOISE | 0 | 0.05 | 0.1 | 0.15 | 0.2 | 0.25 | 0.3 | 0.35 |
| BASELINES | **PROJ**($D_{\text{cx}}$) | **1** | **1** | **1** | 0.61 | 0.14 | 0.04 | 0.02 | 0.01 |
| (NON-NEURAL) | **FAQ**($D_{\text{cx}}$) | **1** | **1** | **1** | **1** | **1** | 0.21 | 0.01 | 0.01 |
| BASELINES | FGNN **PROJ** | 1 | 1 | 0.73 | 0.28 | 0.10 | 0.04 | 0.02 | 0.01 |
| (NEURAL) | FGNN **FAQ** | 1 | 1 | 1 | 1 | 0.95 | 0.14 | 0.01 | 0.01 |
| CHAINING | CHFGNN **PROJ** | 1 | 1 | 0.94 | 0.83 | 0.68 | 0.37 | 0.02 | 0.01 |
| CHAINING | CHFGNN **FAQ** | 1 | 1 | 1 | 1 | **0.99** | **0.62** | 0.01 | 0.01 |

| REGULAR RANDOM GRAPHS WITH DEGREE 10 | | | | | | |
|---|---|---|---|---|---|---|
| REGULAR (**ACC**) | NOISE | 0 | 0.05 | 0.1 | 0.15 | 0.2 |
| BASELINE | **FAQ**($D_{\text{cx}}$) | 0.002 | 0.003 | 0.003 | 0.002 | 0.003 |
| BASELINES | FGNN **PROJ** | 1 | 0.31 | 0.03 | 0.005 | 0.003 |
| (NEURAL) | FGNN **FAQ** | 1 | **0.95** | 0.10 | 0.005 | 0.002 |
| CHAINING | CHFGNN **PROJ** | 1 | **0.95** | 0.54 | 0.009 | 0.003 |
| CHAINING | CHFGNN **FAQ** | 1 | **0.96** | **0.56** | 0.008 | 0.002 |

only 13% for the non-neural **FAQ** baseline and 24% for single-step FGNNs. Note that $p_{\text{noise}} = 0.2$ corresponds to the setting of Yu et al. (2023) where none of the GNN-based methods achieve positive accuracy. Notably, our FGNN architecture alone (without chaining) already outperforms the neural baselines from Yu et al. (2023), demonstrating the importance of architectural expressiveness.

**Regular graphs: a particularly challenging case.** Regular graphs present a unique challenge where standard approaches fail. The uninformative barycenter matrix $D_{\text{cx}} = \frac{1}{n}\mathbf{1}\mathbf{1}^T$ is one of the solution of the convex relaxation (5), giving **FAQ** no useful initialization. Similarly, MPNNs cannot distinguish between nodes in regular graphs Xu et al. (2018), making them ineffective for this task. Table 2 shows that only our FGNN architecture achieves meaningful performance on regular graphs. Our chained FGNNs gets 56% accuracy at $p_{\text{noise}} = 0.1$ while all other methods essentially fail. This demonstrates the critical importance of both architectural expressiveness and iterative refinement for challenging graph alignment scenarios.

Table 3: Accuracy (**acc**) for sparse Erdős-Rényi graphs as a function of the number ($L+1$) of trained FGNNs and in parentheses the gain due to looping ($N_{\text{loop}} = 60$ - $N_{\text{loop}} = L + 1$). Last line: number of loops for chained FGNNs as a function of the noise $p_{\text{noise}}$ to get optimal **nce**.

| NOISE | 0.15 | 0.2 | 0.25 | 0.3 | 0.35 |
|---|---|---|---|---|---|
| L+1=2 | 0.28 (+0.02) | 0.15 (+0.02) | 0.08 (+0.01) | 0.04 (+0.01) | 0.02 (+0.00) |
| L+1=6 | 0.59 (+0.01) | 0.43 (+0.06) | 0.21 (+0.11) | 0.07 (+0.05) | 0.03 (+0.01) |
| L+1=10 | 0.85 (+0.01) | 0.72 (+0.06) | 0.43 (+0.13) | 0.11 (+0.19) | 0.04 (+0.03) |
| L+1=14 | 0.91 (+0.00) | 0.86 (+0.01) | 0.57 (+0.13) | 0.16 (+0.21) | 0.04 (+0.04) |
| L+1=16 | 0.92 (+0.00) | 0.88 (+0.01) | 0.61 (+0.12) | 0.19 (+0.26) | 0.04 (+0.04) |
| #LOOP | 15 | 23 | 88 | 91 | 73 |

## 5.2 LOOPING: ENHANCED INFERENCE WITHOUT ADDITIONAL TRAINING

The chaining procedure trains $L+1$ FGNNs: $f, g^{(1)}, \ldots, g^{(L)}$, with performance typically improving as $L$ increases, see Table 3. Since $g^{(L)}$ refines partial solutions, looping where the final FGNN $g^{(L)}$ is repeatedly applied with the ranking function $r$ (Section 3) for up to $N_{\text{loop}}$ iterations progressively improves accuracy. This gain is shown in parentheses in Table 3 corresponding to the increase in accuracy between no looping, i.e. $N_{\text{loop}} = L+1$ and looping with $N_{\text{loop}} = 60$. We see substantial gain with looping particularly on harder instances ($p_{\text{noise}} = 0.25$ or $0.3$), while incurring minimal computational overhead. In order to get the better results in Table 2, we used looping as long as **nce** continues to improve, capped at $N_{\text{loop}} = 100$ iterations. We see in the last line of Table 3 the average number of loops performed before **nce** plateaus. The results indicate that more difficult problems generally require more iterations, whereas extremely challenging cases ($p_{\text{noise}} = 0.35$) yield no further improvements and thus converge with fewer loops.

## 5.3 TRAINING STRATEGY: OPTIMAL NOISE SELECTION

A key finding is the importance of training noise selection. Figure 3 shows that intermediate noise levels (around $p_{\text{noise}} = 0.22$ for sparse graphs) yield the best generalization. Training on too-easy instances produces models that fail to generalize to harder cases, while training on too-hard instances yields suboptimal performance on easier problems. This "sweet spot" balances challenge and learnability, enabling robust feature learning. All results in Tables 2 use models trained at these optimal noise levels, tuned separately for each graph family.

## 5.4 COMPUTATIONAL EFFICIENCY ANALYSIS

A fair comparison of running times between **FAQ**($D_{\text{cx}}$) and our chained GNN procedure is challenging, so we focus on inference complexity. While our method requires an initial GPU-based training phase, this is assumed to be completed before solving new instances.

For **FAQ**($D_{\text{cx}}$), each gradient step involves solving a linear assignment problem ($O(n^3)$), and total runtime depends on the number of gradient ascent iterations.

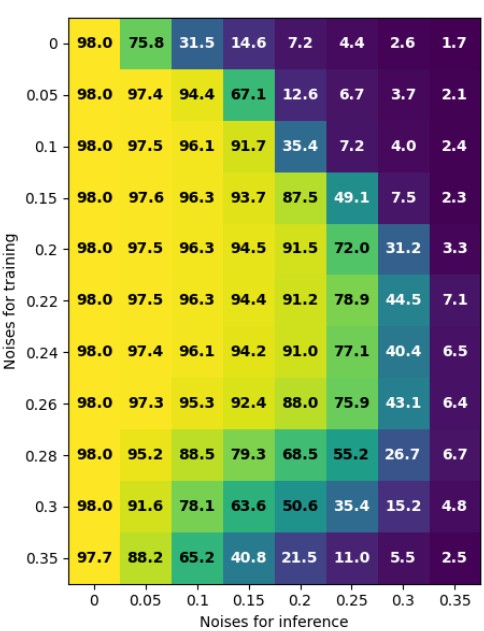

Figure 3: Each line corresponds to chained FGNNs trained at a given level of noise and evaluated across all different level of noises. Performances are **acc** (in %) for sparse Erdős-Rényi graphs with **Proj** as post-processing.

Our chaining procedure has two main costs as $n$ grows: $(i)$ an $n \times n$ matrix multiplication in the graph layer, scaling as $O(n^3)$ but efficient on GPUs, with memory as the main bottleneck; and $(ii)$ computing ranks via a projection **Proj** of the similarity matrix in each iteration, an $O(n^3)$ CPU operation. Table 4 reports the average number of gradient ascent iterations in **FAQ**, starting from either $D_{\text{cx}}$ or the similarity matrix produced by our chained FGNN. The iteration count is substantially lower with the chained FGNN, especially on hard instances ($p_{\text{noise}} \in [0.15, 0.3]$), indicating that the similarity matrix from chaining provides a more accurate initialization than $D_{\text{cx}}$.

Table 4: Average number of gradient projections (**Proj**) in the Frank-Wolfe algorithm **FAQ**, with initialization from either $D_{\text{cx}}$ or the similarity matrix produced by the chained FGNNs.

| ER 4 | NOISE | 0 | 0.05 | 0.1 | 0.15 | 0.2 | 0.25 | 0.3 | 0.35 |
|---|---|---|---|---|---|---|---|---|---|
| **FAQ**($D_{\text{cx}}$) | #ITER | 3.0 | 3.2 | 6.2 | 15.6 | 31.4 | 25.8 | 24.2 | 25.6 |
| CHFGNN **FAQ** | #ITER | 2.0 | 2.1 | 2.8 | 4.1 | 6.5 | 8.3 | 15.1 | 19.7 |

Table 5: Accuracy (in percent) and percent of common edges on the Yeast PPI networks. ChFGNN ER4 is our model trained on Erdős-Rényi graphs with average degree 4, while ChFGNN is trained on the pairs obtained with the three first networks.

| YEAST PPI NETWORKS (ACC / PERCENT OF COMMON EDGES ) | | | | | |
|---|---|---|---|---|---|
| METHOD | 5% CONF | 10% CONF | 15% CONF | 20% CONF | 25% CONF |
| **FAQ**($D_{\text{cx}}$) | 84.2 / 100 | 82.6 / 99.9 | 78.0 / 99.6 | 77.0 / 99.6 | 76.1 / 99.8 |
| CHFGNN ER4 | 80.3 / 99.7 | 75.3 / 99.6 | 67.2 / 99.1 | 63.1 / 98.7 | 53.1 / 97.1 |
| CHFGNN | TRAINING | TRAINING | TRAINING | 72.2 / 99.7 | 69.8 / 99.6 |

## 5.5 RESULTS ON REAL GRAPHS

We evaluate on three real-world datasets from different domains: biology, social networks, and road networks (see details in Section A.3). **Yeast** (Vijayan & Milenkovic, 2018) is a protein–protein interaction (PPI) network with 1,004 proteins and 8,323 trusted interactions. Five noisy variants are created by adding $q\%$ low-confidence edges ($q \in \{5, 10, 15, 20, 25\}$). The base graph is always an induced subgraph of each variant, so the maximum number of common edges is fixed at 8,323. Because the true node correspondence is known, we evaluate alignment quality using accuracy **acc** and normalized number of common edges $\frac{\text{nce}(\pi^{\text{ALG}})}{\text{nce}(\pi^{A \to B})}$.

We first tested chained FGNNs trained on sparse Erdős–Rényi graphs (Section 5.1) to assess transferability. We then trained chained FGNNs on graph pairs constructed from the base network and noisy variants with $q \in \{5, 10, 15\}$, and tested on $q \in \{20, 25\}$. As shown in Table 5, all methods recover nearly the maximum number of common edges (within 3%), but this does not necessarily translate into high node-level accuracy. The base graph has a large automorphism group, so many node permutations preserve edges, and adding edges only worsens identifiability. Thus, although this dataset is a common benchmark, **nce** is the more reliable metric, and **FAQ** is already near-optimal.

To obtain more challenging benchmarks, we also applied the edge-addition–removal noise model (Section 2.3) to the yeast PPI network with $q = 25\%$, the **ca-netscience** coauthorship network (Newman, 2006), and the **inf-euroroad** road network (Šubelj & Bajec, 2011). We evaluated both transferred FGNNs and models specifically trained on these datasets. Table 6 shows that trained ChFGNNs achieve the best performance, with **nce** improving by only about 2% under high noise. As before, node accuracy may not correlate strongly with **nce** due to inherent graph symmetries.

Table 6: Accuracy (in percent) and number of common edges (**nce**) on noisy versions of real-world networks. Each network is corrupted by adding noise at different levels. ChFGNN ER4 is trained on Erdős-Rényi graphs, while ChFGNN is trained on the specific network and noise level. In bold if gain in **nce** is larger than 2%.

| REAL-WORLD NETWORKS WITH ADDED NOISE (ACC / NCE) | | | | | |
|---|---|---|---|---|---|
| METHOD | YEAST25LC | | CA-NETSCIENCE | | INF-EUROROAD | |
| | 5% | 10% | 10% | 20% | 10% | 20% |
| **FAQ**($D_{\text{cx}}$) | 49.8 / 7660 | 44.7 / 7245 | 65.2 / 822 | 45.6 / 687 | 55.8 / 1170 | 10.9 / 940 |
| CHFGNN ER4 | 47.6 / 7693 | 42.3 / 7297 | 63.5 / 818 | 44.1 / 688 | 40.0 / 1111 | 7.5 / **970** |
| CHFGNN | 54.1 / 7732 | 51.3 / **7404** | 65.4 / 824 | 57.0 / **724** | 63.5 / **1213** | 15.4 / **963** |

## 6 CONCLUSION

In summary, we introduced a chaining procedure with GNNs for tackling the combinatorial graph alignment problem, achieving substantial performance gains and compatibility with existing solvers. We further proposed a challenging benchmark of correlated regular graphs, for which no competing algorithms are known. Our method extends naturally to the seeded variant of GAP, and we anticipate that the chaining framework may generalize to other combinatorial problems, offering promising directions for future research.

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

# A  APPENDIX

## CONTENTS

## A.1  CORRELATED RANDOM GRAPHS AND PREVIOUS RECENT RESULTS

In this section, we present the mathematical details for the various correlated random graphs model used in this paper.

**Bernoulli graphs.**  We start with the model considered in (Lyzinski et al., 2015). Given $n$ the number of nodes, $\rho \in [0, 1]$ and a symmetric hollow matrix $\Lambda \in [0, 1]^{n \times n}$, define $\mathcal{E} = \{\{i, j\}, i \in [n], j \in [n], i \neq j\}$. Two random graphs $G_A = (V_A, E_A)$ and $G_B = (V_B, E_B)$ are $\rho$-correlated Bernoulli($\Lambda$) distributed, if for all $\{i, j\} \in \mathcal{E}$, the random variables (matrix entries) $A_{ij}$ and $B_{ij}$ are such that $B_{ij} \sim$ Bernoulli($\Lambda_{ij}$) independently drawn and then conditioning on $B$, we have $A_{ij} \sim$ Bernoulli($\rho B_{ij} + (1 - \rho)\Lambda_{ij}$) independently drawn. Note that the marginal distribution of $A$ and $B$ are Bernoulli($\rho \Lambda + (1 - \rho)\Lambda$) distributed, i.e. the laws of $A$ and $B$ are the same (but correlated).

In our experiments in Sections A.10.1 and A.10.2, we consider the same case as in (Lyzinski et al., 2015): $n = 150$ vertices, the entries of the matrix $\Lambda$ are i.i.d. uniform in $[\alpha, 1 - \alpha]$ with $\alpha = 0.1$, and we vary $\rho$.

**Erdős-Rényi graphs.**  The Erdős-Rényi model is a special case of the Bernoulli model where $\Lambda$ is the matrix with all entries equal to $\lambda$. To be consistent with the main notation, we define $p_{\text{noise}} = (1 - \lambda)(1 - \rho)$ where $\rho$ was the correlation above and $\lambda = d/n$ where $d$ is the average

Table 7: Statistics of synthetic datasets.

| name | average degree | number of nodes | useed for comparison with | sizes of train/val |
|------|---------|---------|---------|---------|
| Bernoulli | 70 | 150 | **FAQ**($D_{cx}$) (Lyzinski et al., 2015) | 2000/200 |
| Sparse Erdős-Rényi (ER 4) | 4 | 500 | MPNN (Yu et al., 2023) | 200/100 |
| Dense Erdős-Rényi (ER 80) | 80 | 500 | MPNN (Yu et al., 2023) | 200/100 |
| Large Erdős-Rényi | 3 | 1000 | Bayesian message passing (Muratori & Semerjian, 2024) | 200/100 |
| Regular | 10 | 500 | new | 200/100 |

degree of the graph. Hence the random graphs $G_A$ and $G_B$ are correlated Erdős-Rényi graphs when $\mathbb{P}(A_{i,j} = B_{i,j} = 1) = \frac{d}{n}(1 - p_{\text{noise}})$ and $\mathbb{P}(A_{i,j} = 0, B_{i,j} = 1) = \mathbb{P}(A_{i,j} = 1, B_{i,j} = 0) = \frac{d}{n}p_{\text{noise}}$.

**Regular graphs.** In this case, we first generate $G_A$ as a uniform regular graph with degree $d$ and then we generate $G_B$ by applying edgeswap to $G_A$: if $\{i, j\}$ and $\{k, \ell\}$ are two edges of $G_A$ then we swap them to $\{i, \ell\}$ and $\{k, j\}$ with probability $p_{\text{noise}}$.

The problem of graph alignment for correlated Erdős Rényi random graphs has been studied empirically with Message Passing GNN (MPNN) in (Yu et al., 2023) when a seed of matched vertices is given in addition to the 2 graphs. We are reproducing their results taken from `https://github.com/Leron33/SeedGNN` corresponding to Figure 6 in (Yu et al., 2023). SeedGNN refers to (Yu et al., 2023), PGM to (Kazemi et al., 2015), SGM to (Fishkind et al., 2019) and MGCN to(Chen et al., 2020)

Table 8: Accuracy (%) on sparse Erdős Rényi random graphs with average degree 4 and noise 0.2 as a function of the seed

| Fraction of Seeds | 0% | 2% | 4% | 6% | 8% | 10% | 12% | 14% | 16% | 18% | 20% |
|------|------|------|------|------|------|------|------|------|------|------|------|
| SeedGNN | 0.3 | 15.1 | 47.4 | 82.8 | 96.0 | 96.6 | 97.0 | 97.6 | 97.6 | 97.6 | 97.6 |
| PGM | 0.2 | 2.3 | 6.1 | 16.3 | 31.6 | 54.5 | 73.3 | 79.2 | 86.3 | 88.9 | 92.7 |
| SGM | 0.3 | 3.6 | 8.9 | 13.8 | 22.3 | 36.3 | 54.5 | 67.3 | 84.4 | 89.6 | 91.6 |
| MGCN | 0.1 | 2.0 | 4.0 | 6.7 | 8.4 | 11.1 | 12.4 | 14.0 | 16.3 | 18.9 | 20.5 |

Looking at the results from Section 5, we see that:

- for sparse Erdős Rényi random graphs (Table 8) with no seed and a noise of 0.2, the accuracy for **FAQ**($D_{cx}$) is 73% and for our chained FGNNs 93%.

- for dense Erdős Rényi random graphs (Table 9) with no seed and a noise of 0.2, the accuracy for **FAQ**($D_{cx}$) is 100% and for our chained FGNNs 99%.

Table 9: Accuracy (%) on dense Erdős Rényi random graphs with average degree 80 and noise 0.2 as a function of the seed

| Fraction of Seeds | 0% | 0.5% | 1% | 1.5% | 2% | 2.5% | 3% | 3.5% | 4% | 4.5% | 5% |
|------|------|------|------|------|------|------|------|------|------|------|------|
| SeedGNN | 0.1 | 0.7 | 91.4 | 100 | 100 | 100 | 100 | 100 | 100 | 100 | 100 |
| PGM | 0.1 | 0.6 | 1.8 | 4.3 | 19.3 | 51.2 | 96.6 | 100 | 100 | 100 | 100 |
| SGM | 0.2 | 1.5 | 85.8 | 100 | 100 | 100 | 100 | 100 | 100 | 100 | 100 |
| MGCN | 0.1 | 0.7 | 1.5 | 1.9 | 3.7 | 5.2 | 6.9 | 8.0 | 10.9 | 12.3 | 13.7 |

## A.2 TIME–PERFORMANCE TRADE-OFF

For iterative algorithms, the computation time can be controlled by adjusting the number of iterations. This applies to gradient-based methods such as the Frank–Wolfe algorithm, used to compute either the convex relaxation $D_{cx}$ or the indefinite relaxation **FAQ**. The recently proposed FUGAL

algorithm (Bommakanti et al., 2024) is also iterative. Finally, our chained FGNNs naturally define an iterative procedure, where we may bound the number of chaining steps. Figure 4 displays the resulting Pareto curves, showing the trade-off between performance and runtime for each method on sparse Erdős–Rényi graphs with noise level $p_{\text{noise}} = 0.2$.

For FUGAL, we used the authors' implementation (available at `https://github.com/idea-iitd/Fugal`), specifically the `predict_alignment` routine with hyperparameter `mu = 1`. For **FAQ**($J$), we relied on the SciPy implementation `scipy.optimize.quadratic_assignment(method='faq')`. For **FAQ**($D_{\text{cx}}$), we used our own Frank–Wolfe implementation to compute $D_{\text{cx}}$ before passing it to **FAQ**.

We emphasize that **FAQ** and FUGAL run on CPU, whereas our chained FGNNs run on GPU. Runtimes correspond to computing the number of common edges (**nce**) over 100 graph pairs of size $n = 500$. As shown in the figure, our chained FGNNs achieve better performance at lower computation time compared to both **FAQ** and FUGAL.

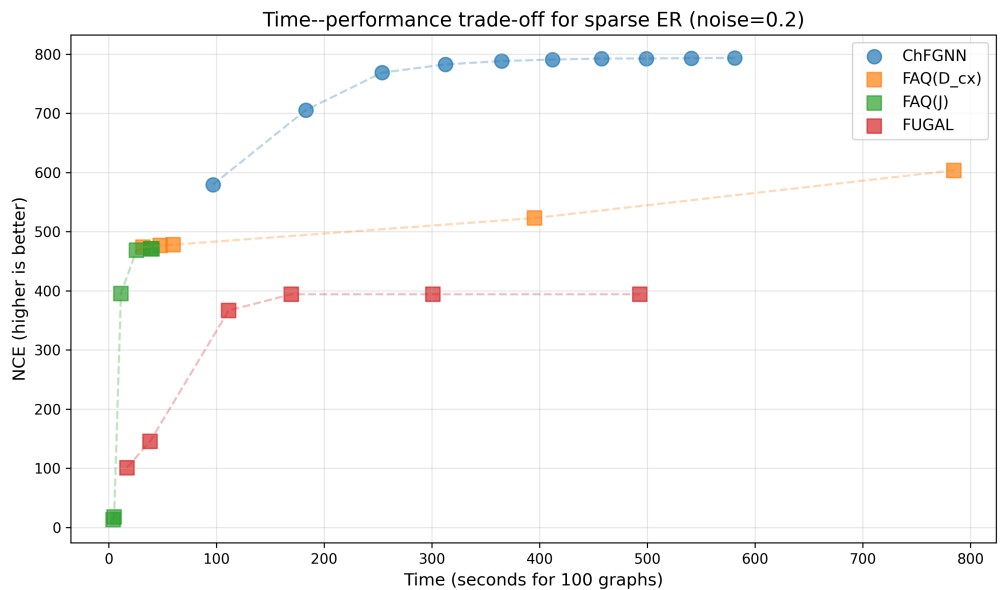

Figure 4: Number of common edges (**nce**) recovered in sparse Erdős–Rényi graphs as a function of the computation time allocated to each algorithm: FUGAL, **FAQ**($J$), **FAQ**($D_{\text{cx}}$), and our chained FGNNs (ChFGNN).

### A.3 REAL GRAPHS AND PREVIOUS RECENT RESULTS

The real-world networks in Section 5.5 are standard benchmarks for graph alignment. We apply the same noising procedure as for Erdős–Rényi graphs (described above), using the original graph's average degree $d$. Table 10 gives the sizes of the training and validation sets used for training our chained FGNNs.

Table 10: Summary statistics of the real-world graphs.

| Dataset | # Nodes | # Edges | Avg. Degree | size train/valid |
|---|---|---|---|---|
| Yeast PPI (Vijayan & Milenkovic, 2018) | 1,004 | 8,323 | 16.58 | 20/20 |
| ca-netscience(Newman, 2006) | 379 | 914 | 4.82 | 200/20 |
| inf-euroroad (Šubelj & Bajec, 2011) | 1,174 | 1,417 | 2.41 | 20/5 |

Table 11 reports results for SGWL (Xu et al., 2019) and FUGAL (Bommakanti et al., 2024). SGWL results come from the original paper; FUGAL results were obtained using the authors' code (available at `https://github.com/idea-iitd/Fugal`). Our FUGAL performance matches (Bommakanti et al., 2024), but our **FAQ** results do not. We attribute the discrepancy to

poor initialization: using the uninformative barycenter $J = \frac{1}{n}\mathbf{1}\mathbf{1}^\top$ reproduces the degraded **FAQ** performance reported in (Bommakanti et al., 2024).

Table 11: Accuracy (**acc**) and pnumber of common edges (**nce**) on the Yeast PPI networks. ChFGNN ER4 is our model trained on Erdős-Rényi graphs with average degree 4, while ChFGNN is trained on the pairs obtained with the three first networks.

| YEAST PPI NETWORKS (ACC / NCE ) | | | | | |
|---|---|---|---|---|---|
| METHOD | 5% CONF | 10% CONF | 15% CONF | 20% CONF | 25% CONF |
| **FAQ**($J$) | 37.5 / 7383 | 34.4 / 7245 | 29.1 / 6807 | 23.9 / 6689 | 36.4 / 7383 |
| SGWL | 83.6 / – | – / – | 66.6 / – | – / – | 58.8 / – |
| FUGAL | 83.0 / 8311 | 77.7 / 8231 | 74.3 / 8172 | 70.9 / 8148 | 68.6 / 8095 |
| **FAQ**($D_{\text{cx}}$) | 84.2 / 8323 | 82.6 / 8317 | 78.0 / 8289 | 77.0 / 8294 | 76.1 / 8306 |
| CHFGNN ER4 | 80.3 / 8300 | 75.3 / 8288 | 67.2 / 8252 | 63.1 / 8213 | 53.1 / 8080 |
| CHFGNN | TRAINING | TRAINING | TRAINING | 72.2 / 8300 | 69.8 / 8291 |

On noisy real datasets (Table 12), FUGAL never matches **FAQ**, contrary to the claims in (Bommakanti et al., 2024). To compute the maximum number of common edges, we run **FAQ** initialized with the true permutation (prior to noising).

Table 12: Accuracy (**acc**) and number of common edges (**nce**) on noisy versions of real-world networks. Each network is corrupted by adding noise at different levels. ChFGNN ER4 is trained on Erdős-Rényi graphs, while ChFGNN is trained on the specific network and noise level.In bold if gain in **nce** is larger than 2%.

| REAL-WORLD NETWORKS WITH ADDED NOISE (ACC / NCE) | | | | | | |
|---|---|---|---|---|---|---|
| METHOD | YEAST25LC | | CA-NETSCIENCE | | INF-EUROROAD | |
| | 5% | 10% | 10% | 20% | 10% | 20% |
| FUGAL | 53.1 / 7480 | 44.6 / 7035 | 60.3 / 794 | 37.7 / 629 | 18.3 / 818 | 2.9 / 714 |
| **FAQ**($D_{\text{cx}}$) | 49.8 / 7660 | 44.7 / 7245 | 65.2 / 822 | 45.6 / 687 | 55.8 / 1170 | 10.9 / 940 |
| CHFGNN ER4 | 47.6 / 7693 | 42.3 / 7297 | 63.5 / 818 | 44.1 / 688 | 40.0 / 1111 | 7.5 / **970** |
| CHFGNN | 54.1 / 7732 | 51.3 / **7404** | 65.4 / 824 | 57.0 / **724** | 63.5 / **1213** | 15.4 / **963** |
| MAX NCE | 7909 | 7498 | 826 | 730 | 1272 | 1137 |

FUGAL is substantially faster than **FAQ**($D_{\text{cx}}$), though we did not perform a detailed timing study. We lack an efficient implementation of the Frank–Wolfe solver required for the convex-relaxation initialization $D_{\text{cx}}$, and our implementation prioritizes correctness over speed. The subsequent **FAQ** step uses SciPy's efficient routine `quadratic_assignment`. We expect that **FAQ**($D_{\text{cx}}$) could be made significantly faster with an optimized implementation.

### A.4    GNN ARCHITECTURE AND EXPRESSIVENESS

The choice of a more expressive architecture is crucial for our approach. The success of the chaining procedure critically depends on producing a high-quality initial similarity matrix $S^{A \to B}(0)$ to bootstrap the iterative refinement process. Standard MPNNs, which aggregate only local neighborhood information, would produce similarity matrices based on limited local features—insufficient for capturing the global structural patterns needed for effective graph alignment. This limitation has been observed in prior work: Nowak et al. (2018) implemented a similar initial step using MPNNs with limited success, while Azizian & Lelarge (2021) demonstrated the superiority of Folklore-type GNNs for this task. As we show in Section 5, combining our expressive architecture with the chaining procedure yields substantial performance improvements over single-step approaches.

**Core architecture: Folklore-inspired residual layers.** Our GNN's main building block is a residual layer that processes hidden states for all node pairs $(h_{i \to j}^t)_{i,j} \in \mathbb{R}^{n \times n \times d}$, producing updated

states $(h_{i \to j}^{t+1})_{i,j} \in \mathbb{R}^{n \times n \times d}$:

$$h_{i \to j}^{t+1} = h_{i \to j}^{t} + m_1 \left( h_{i \to j}^{t}, \sum_{\ell} h_{i \to \ell}^{t} \odot m_0(h_{\ell \to j}^{t}) \right), \tag{12}$$

where $m_0 : \mathbb{R}^d \to \mathbb{R}^d$ and $m_1 : \mathbb{R}^{2d} \to \mathbb{R}^d$ are multilayer perceptrons (MLPs) with graph normalization layers, and $\odot$ denotes component-wise multiplication.

This design incorporates several improvements over the original Folklore-type GNN (Maron et al., 2019):

- **Residual connections**: The skip connection $h_{i \to j}^{t} + (\cdot)$ enables deeper networks and more stable training.
- **Graph normalization**: Inspired by Cai et al. (2021), this ensures well-behaved tensor magnitudes across different graph sizes.
- **Simplified architecture**: We use only one MLP in the component-wise multiplication, reducing memory requirements while maintaining expressiveness.

**Input and output transformations.** The complete architecture consists of three main components:

1. **Input embedding**: The adjacency matrix $A \in \{0,1\}^{n \times n}$ is embedded into the initial hidden state $h_{i \to j}^{0} \in \mathbb{R}^{n \times n \times d}$ using a learned embedding layer that encodes edge presence/absence.
2. **Residual processing**: Multiple residual layers (12) transform the node-pair representations, capturing complex structural relationships.
3. **Node feature extraction**: The final tensor $h_{i \to j}^{k} \in \mathbb{R}^{n \times n \times d}$ is converted to node features $\mathbb{R}^{n \times d}$ via max-pooling over the first dimension: $\text{node}_i = \max_j h_{i \to j}^{k}$.

**Ranking integration for chained networks.** The networks $g^{(1)}, g^{(2)}, \ldots$ must incorporate ranking information in addition to graph structure. We achieve this through learned positional encodings that map each node's rank to a $d$-dimensional vector. These rank embeddings are concatenated with the node features from the max-pooling layer, allowing the network to leverage both structural and ranking information when computing enhanced similarities.

This architecture provides the expressiveness needed to capture global graph properties while remaining trainable through the sequential training procedure described in Section 3.2. While scalability remains a limitation for very large graphs, the architecture proves highly effective for the graph sizes considered in our experiments (up to 1000 nodes).

### A.5 TECHNICAL DETAILS FOR THE GNN ARCHITECTURE AND TRAINING

By default, we use MLP for the functions $m_0$ and $m_1$ in (12) with 2 hidden layers of dimension 256. In all our experiments, we take a GNN with 2 residual layers. We used Adm optimizer with a learning rate of $1e-4$ and the scheduler ReduceLROnPlateau from PyTorch with a patience parameter of 3.

For **Proj**, we use the function **linear_sum_assignment** from **scipy.optimize** and for **FAQ**, we use the function **quadratic_assignment** from the same library. SciPy is a set of open source (BSD licensed) scientific and numerical tools for Python. In order to compute $D_{\text{cx}}$ solving (5), we implemented the Frank-Wolfe algorithm.

For the training and inference, we used Nvidia RTX8000 48GB and Nvidia A100 80GB. For graphs of size 500, we train on 200 graphs and validate on 100 graphs for 100 epochs. We run for $L = 15$ steps of chaining (obtaining 16 trained FGNNs: $f, g^{(1)}, \ldots, g^{(15)}$). The PyTorch code is available as a supplementary material.

### A.6 MORE RELATED WORK

Supervised learning approach of the graph matching problem has been greatly studied in the computer vision literature (Wang et al., 2021), (Rolínek et al., 2020), (Zanfir & Sminchisescu, 2018),

(Gao et al., 2021), (Yu et al., 2021), (Jiang et al., 2022). (Fey et al., 2020) is closely related to our work and proposes a two-stage architecture similar to our chaining procedure with MPNNs. The first stage is the same as our first step but with a MPNN instead of our FGNN. Then the authors propose a differentiable, iterative refinement strategy to reach a consensus of matched nodes. All these works assume that non-topological node features are available and informative. This is a setting favorable to GNNs as node-based GNN is effective in learning how to extract useful node representations from high-quality non-topological node features. In contrast, we focus on the pure combinatorial problem where no side information is available. In (Li et al., 2019), graph matching networks take a pair of graphs as input and compute a similarity score between them. This algorithm can be used to compute the value of the graph matching (1) but does not give the optimal permutation $\pi^{A \to B}$ between the two graphs which is the main focus of our work.

Regarding benchmarks for the GAP, we are not aware of any pubicly available dataset. The GAP can be seen as a particular version of the QAP and some algorithms designed for the GAP can be used for QAP instances (i.e. with weighted adjacency matrices). This is the case for the convex and indefinite relaxations presented in Section 2.2 which can be used with real-valued matrices. In particular, (Lyzinski et al., 2015) shows very good performances of **FAQ** on some QAP instances from (Burkard et al., 1997). These instances are small (from 12 to 40 nodes) with full (integer-valued) matrices. They are very far from the distribution of correlated random graphs used for training in our work and we do not expect good performaces for such out-of-distribution instances for any supervised learning algorithm.

### A.7 MODEL-BASED VERSUS SIMULATION-BASED ALGORITHMS

As explained in Section 2.3, we train and evaluate our supervised learning algorithms on correlated random graphs. This choice connects our work to a rich theoretical literature on the correlated Erdős-Rényi random graph ensemble, which has been extensively studied from an information-theoretic perspective.

#### A.7.1 THEORETICAL FOUNDATIONS AND LIMITS

The theoretical analysis begins with Cullina & Kiyavash (2016), which establishes the information-theoretic limit for exact recovery of $\pi^\star$ as the number of nodes $n$ tends to infinity. In the sparse regime, where the average degree $d$ remains constant as $n \to \infty$, exact recovery becomes impossible. Subsequent work by Ganassali et al. (2021) and Ding & Du (2023) demonstrates that partial recovery of $\pi^\star$ is only possible when $p_{\text{noise}} < 1 - d^{-1}$.

For the correlated Erdős-Rényi ensemble, the joint probability distribution is given by:

$$\mathbb{P}(G_A, G_B) = \left( \frac{(1 - p_{\text{noise}})(n^2 - d(1 + p_{\text{noise}}))}{d p_{\text{noise}}^2} \right)^{e(G_A \wedge G_B)},$$

where $e(G_A \wedge G_B) = \sum_{i<j} A_{ij} B_{ij}$ counts the common edges between graphs $G_A$ and $G_B$. This distribution reveals a crucial insight: **the maximum a posteriori estimator of $\pi^*$ given $G_A$ and the permuted graph $G_B'$ is exactly a solution of the GAP on the $(G_A, G_B')$ instance**.

#### A.7.2 MODEL-BASED APPROACHES: ACHIEVEMENTS AND LIMITATIONS

Recent theoretical advances have produced efficient polynomial-time algorithms (Ding et al., 2021; Fan et al., 2023; Ding & Li, 2023; Ganassali et al., 2024a; Piccioli et al., 2022) that approximate the probability distribution by exploiting structural properties like the local tree-like nature of sparse random graphs. These algorithms achieve partial recovery (positive accuracy) when $p_{\text{noise}}$ is sufficiently small, though well below the information-theoretic threshold of $1 - d^{-1}$.

However, a fundamental **algorithmic threshold** appears to exist. Recent work (Mao et al., 2023; Ganassali et al., 2024b) suggests that no efficient algorithm can succeed for $p_{\text{noise}} > p_{\text{algo}} = 1 - \sqrt{\alpha} \approx 0.419$, where $\alpha$ is Otter's constant, even when the average degree $d$ is large.

While these model-based algorithms provide theoretical guarantees for correlated Erdős-Rényi graphs, they suffer from significant practical limitations:

- **Narrow applicability**: Designed specifically for the correlated Erdős-Rényi model with no guarantees outside this distribution

- **Computational complexity**: Despite polynomial-time guarantees, running times are often impractical for real applications

- **Limited scalability**: Most implementations prioritize mathematical rigor over computational efficiency

Muratori & Semerjian (2024) represents a notable exception, focusing on making message-passing algorithms (Ganassali et al., 2024a; Piccioli et al., 2022) more scalable while maintaining theoretical guarantees.

### A.7.3 FAQ: AN EMPIRICAL SURPRISE

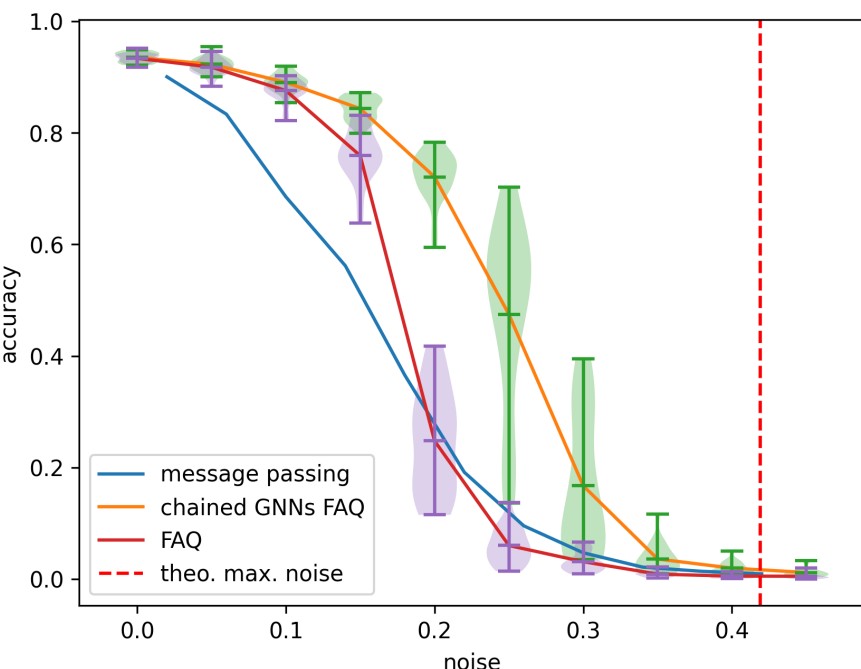

Figure 5: Accuracy **acc** as a function of the noise level for correlated Erdős-Rényi random graphs with size $n = 1000$ and average degree $d = 3$. Chained GNNs were trained at noise level 0.25 and **FAQ** is used as the last step for the inference. The red curve labeled FAQ corresponds to $\textbf{FAQ}(D_{cx})$ and the blue curve labeled message passing are results from (Muratori & Semerjian, 2024). The dashed vertical line corresponds to the theoretical $p_{algo} = 1 - \sqrt{\alpha}$ above which no efficient algorithm is known to succeed.

Remarkably, the **FAQ** algorithm—which was not designed specifically for any random graph model—empirically encounters the same algorithmic barrier predicted by theory. As shown in Figure 5, **FAQ**'s performance degrades sharply near $p_{algo}$, matching the theoretical predictions despite lacking formal guarantees for this setting. **FAQ** only underperforms compared to specialized message-passing methods Muratori & Semerjian (2024) when $p_{noise}$ approaches $p_{algo}$.

This empirical observation suggests that **FAQ**, through its continuous relaxation approach, implicitly captures fundamental structural properties of the graph alignment problem that transcend specific random graph models.

### A.7.4 OUR SIMULATION-BASED APPROACH

To circumvent the computational challenges of maximizing the exact posterior (which, as shown above, corresponds exactly to solving the GAP), we adopt a **simulation-based approach**. Rather than deriving model-specific algorithms, we:

1. **Sample training data**: Generate pairs of graphs $(G_A, G_B)$ with known alignment permutations $\pi^\star$

2. **Learn mappings**: Train neural networks to map graph pairs to similarity matrices $S^{A\to B} \in \mathbb{R}^{n\times n}$

3. **Extract solutions**: Convert similarity matrices to permutations via projection or as **FAQ** initialization

**Key Advantages.** Our simulation-based approach offers several advantages over both model-based methods and traditional relaxations:

**Supervised learning with ground truth**: Unlike the convex relaxation (5), we have access to ground truth permutations during training, enabling more informative loss functions.

**Better optimization objective**: Instead of the Frobenius norm used in convex relaxation, we employ cross-entropy loss, which provides more informative gradients for discrete matching problems.

**Generalization potential**: While trained on specific distributions, our learned representations may capture general structural patterns applicable beyond the training distribution.

**Hybrid capability**: Our similarity matrices can initialize traditional solvers like **FAQ**, combining the benefits of learning and optimization approaches.

This simulation-based methodology bridges the gap between theoretical guarantees and practical performance, achieving strong empirical results while maintaining computational tractability.

### A.8 RELATING GAP TO GROMOV-HAUSDORFF, GROMOV-MONGE AND GROMOV-WASSERSTEIN DISTANCES FOR FINITE METRIC SPACES

We consider a simple case of discrete spaces with the same number of elements $n$ and where $A, B \in \mathbb{R}^{n\times n}$ are the distance matrices of two finite metric spaces $(X, d_X)$ and $(Y, d_Y)$, i.e. $A_{ij} = d_X(x_i, x_j)$ and $B_{ij} = d_Y(y_i, y_j)$. Recall that we denote by $\mathcal{S}_n$ the set of permutation matrices and by $\mathcal{D}_n$ the set of doubly stochastic matrices. We also denote by $\mathcal{R}_n$ the set of matrices $R \in \{0,1\}^{n\times n}$ such that $\sum_i R_{ij} \geq 1$ and $\sum_j R_{ij} \geq 1$.

The **Gromov-Hausdorff distance** for finite metric spaces can be written as:

$$\mathrm{GH}_L(A, B) = \min_{R\in\mathcal{R}_n} \max_{i,j,k,\ell} L(A_{ik}, B_{j\ell}) R_{ij} R_{k\ell} \tag{13}$$

where $L(a, b) \geq 0$. It is often desirable to smooth the max operator in (13) to a sum. This can be done by considering the related problem:

$$\mathrm{GM}_L(A, B) = \min_{R\in\mathcal{R}_n} \sum_{i,j,k,\ell} L(A_{ik}, B_{j\ell}) R_{ij} R_{k\ell} \tag{14}$$

Note that for any $R \in \mathcal{R}$, there exists a permutation matrix $P \in \mathcal{S}_n$ such that $R_{ij} \geq P_{ij}$ so that we have: $\sum_{i,j,k,\ell} L(A_{ik}, B_{j\ell}) R_{ij} R_{k\ell} \geq \sum_{i,j,k,\ell} L(A_{ik}, B_{j\ell}) P_{ij} P_{k\ell}$. Therefore, the minimum in (14) is attained at some $R \in \mathcal{S}_n$. In particular, we get:

$$\mathrm{GM}_L(A, B) = \min_{P\in\mathcal{S}_n} \sum_{i,j,k,\ell} L(A_{ik}, B_{j\ell}) P_{ij} P_{k\ell} = \min_{\pi\in\mathcal{S}_n} \sum_{i,j} L\left(A_{ij}, B_{\pi(i)\pi(j)}\right),$$

which is called **Gromov-Monge distance**.

The **Gromov-Wasserstein distance** is a relaxation of the Gromov-Hausdorff distance and is defined in Mémoli (2011):

$$\mathrm{GW}_L(A, B, p, q) = \min_{T\in\mathcal{C}_{p,q}} \sum_{i,j,k,\ell} L(A_{ik}, B_{j\ell}) T_{ij} T_{k\ell}, \tag{15}$$

where $p, q$ are probability distributions on $X$, $Y$ and the minimum is taken over $\mathcal{C}_{p,q} = \{T \in \mathbb{R}_+^{n \times n}, T\mathbf{1} = p, T^T\mathbf{1} = q\}$. Taking $p = q = \mathbf{1}/n$ the uniform distribution, we have $\mathcal{C}_{p,q} = \frac{1}{n}\mathcal{D}_n$ and $\mathrm{GW}_L(A, B, \mathbf{1}/n, \mathbf{1}/n)$ is a relaxed version of $\mathrm{GM}_L(A, B)$. We typically consider $L(a, b) = |a - b|^2$, and then we get:

$$\mathrm{GM}_{L^2}(A, B) = \min_{\pi \in \mathcal{S}_n} \sum_{i,j} (A_{ij} - B_{\pi(i)\pi(j)})^2,$$

and with the simplified notation $\mathrm{GW}_{L^2}(A, B) = \mathrm{GW}_{L^2}(A, B, \mathbf{1}/n, \mathbf{1}/n)$,

$$n^2 \mathrm{GW}_{L^2}(A, B) = \min_{D \in \mathcal{D}_n} \sum_{i,j,k,\ell} (A_{ik} - B_{j\ell})^2 D_{ij} D_{k\ell}$$

$$= \min_{D \in \mathcal{D}_n} \sum_{i,k} A_{ik}^2 + \sum_{j\ell} B_{j\ell}^2 - 2 \sum_{i,j,k,\ell} A_{ik} B_{j\ell} D_{ij} D_{k\ell}$$

$$= \|A\|_F^2 + \|B\|_F^2 - 2 \max_{D \in \mathcal{D}_n} \langle AD, DB \rangle.$$

In the particular case where $A$ and $B$ are semi definite positive matrices, i.e. $A = U^T U$ and $B = V^T V$, we have: $\langle AD, DB \rangle = \|UDV^T\|_F^2$ which is a convex function of $D$ and is always maximized at an extremal point of its constraint polytope $\mathcal{D}_n$. By Birkhoff's theorem, the extremal points of $\mathcal{D}_n$ are permutation matrices. Therefore, we have: $\mathrm{GW}_{L^2}(A, B) = \mathrm{GM}_{L^2}(A, B)$ in this case. Maron & Lipman (2018) shows that a similar result holds for Euclidean distances, when $A_{ij} = \|x_i - x_j\|_2$ and $B_{ij} = \|y_i - y_j\|_2$. Hence, we have:

**Proposition A.1.** *For $A$, $B$ Euclidean distance matrices, the indefinite relaxation* (5) *is tight and solves the GAP* (1)*. In this case, the GAP computes the Gromov-Monge distance and the indefinite relaxation computes the Gromov-Wasserstein distance.*

### A.9 NOTATIONS USED IN TABLES

- **acc FAQ**($D_{\mathrm{cx}}$) means the accuracy of **FAQ** algorithm initialized with $D_{cx}$.
- **acc** ChFGNN **Proj** means the accuracy of our chained FGNNs with **Proj** as the last step.
- **acc** ChFGNN **FAQ** means the accuracy of our chained FGNNs with **FAQ** as the last step.
- **nce FAQ**($D_{\mathrm{cx}}$) means the number of common edges found by **FAQ** algorithm initialized with $D_{cx}$.
- **nce FAQ**($\pi^\star$) means the number of common edges found by **FAQ** algorithm initialized with $\pi^\star$.
- **nce** ChFGNN **Proj** means the number of common edges found by our chained FGNNs with **Proj** as the last step.
- **nce** ChFGNN **FAQ** means the number of common edges found by our chained FGNNs with **FAQ** as the last step.

### A.10 BERNOULLI GRAPHS: GENERALIZATION PROPERTIES FOR CHAINED GNNS

### A.10.1 TRAINING CHAINED GNNS

For the same dataset as in Lyzinski et al. (2015) (see Bernoulli graphs in Section A.1), we plot in Figure 6 the training curves for the chained GNNs for the loss and the accuracy on the training set and the validation set. We do not see any overfitting here as values are similar on both sides. Chaining is very effective in this case: while the first training (brown) corresponding to the mapping $f$ in (10) saturates at an accuracy below $0.1$, the second training (magenta) corresponding to the mapping $g^{(1)}$ in (11) reaches a much higher accuracy because it uses the information about the graph matching contained in the output of the first training $f$. The curves for the remaining trainings are indeed ordered. This is due to the fact that for the training of $g^{(k+1)}$, we initialized it with the weights obtained after the training of $g^{(k)}$ in order to speed up the training. Since we observe a saturation in the learning of the $g^{(k)}$ for $k \geq 2$, we stop the training after half the number of epochs used for $f$ and $g^{(1)}$.

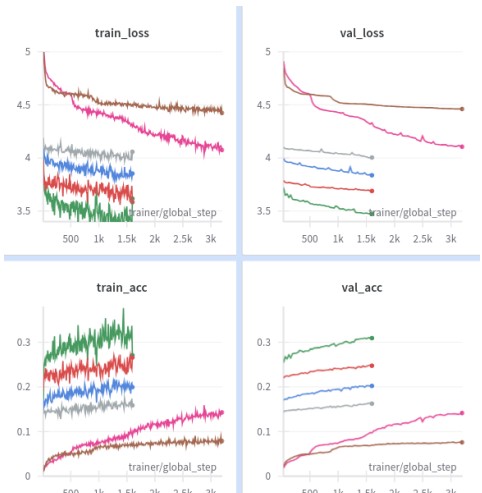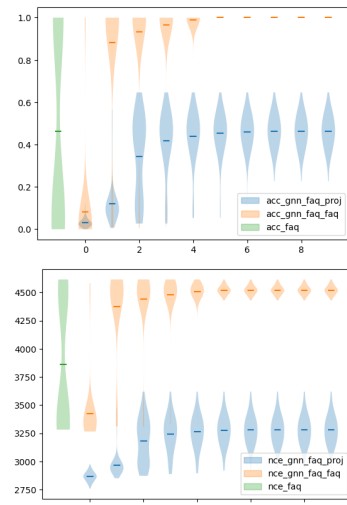

Figure 6: Left: Training chained GNNs. Each color corresponds to a different training and GNN: the first training (brown) reaches an accuracy below 0.1. The second training (magenta) uses as input the output of the first training and get an accuracy $\approx 0.15$. The remaining trainings using the output of the previous training as input and reach higher and higher accuracy.
Right top: **acc** bottom: **nce**. First Violin plot (_**faq** green) for **FAQ**, then all other Violin plots correspond to a different number of iterations $N_{\max} = 0, 1, \ldots, 9$ of the chaining procedure (_**gnn_faq_proj** blue with **Proj** and _**gnn_faq_faq** orange with **FAQ**).

### A.10.2 EFFICIENT INFERENCE FOR CHAINED FGNNS

These results suggest an extreme form of looping: since $g^{(1)}$ allows to improve the accuracy of an initial guess (given by $f$), we can keep only the GNNs $f$ and $g^{(1)}$, and we loop through $g^{(1)}$ for a fixed number of steps $N_{\max}$. Figure 6 gives the accuracy **acc** defined in (3) and the number of common edges **nce** defined in (4) for the inference procedure as a function of the number of iterations $N_{\max}$ made on $g^{(1)}$. We give (in blue) the performances of **Proj**, and (in orange) the performances of **FAQ** applied on the similarity matrix obtained after $L$ loops. We see that the Frank-Wolfe algorithm used in **FAQ** used as the last step of our chaining procedure is crucial to get better performances. Indeed, we need only $N_{\max} = 5$ loops in order to get a perfect accuracy **acc** $= 1$ with **FAQ**.

We also give (in green) the performance of **FAQ** applied on the matrix $D_{\mathrm{cx}}$ (the default option in the **FAQ** algorithm). Indeed, **FAQ**$(D_{\mathrm{cx}})$ is able to find the correct permutation for the graph matching in 13% of the cases and is stuck in a local maxima with a very small (less than 20%) accuracy otherwise. This bimodal behavior is due to the fact that $D_{\mathrm{cx}}$ gives very little information about the correct permutation. In contrast, **the chaining procedure was able to learn a much better initialization than $D_{\mathrm{cx}}$ for FAQ allowing to improve the accuracy from** 50% **to an exact accuracy.**

|  | noise | 0.4 | 0.45 | 0.5 | 0.55 | 0.6 | 0.65 | 0.7 |
|---|---|---|---|---|---|---|---|---|
|  | acc **Proj**$(D_{\mathrm{cx}})$ | 0.3428 | 0.1956 | 0.1209 | 0.0815 | 0.0552 | 0.0411 | 0.0309 |
|  | acc **FAQ**$(D_{\mathrm{cx}})$ | 1.0 | 0.9954 | 0.9531 | 0.6910 | 0.2621 | 0.0959 | 0.0225 |
|  | nce **Proj**$(D_{\mathrm{cx}})$ | 3147.7 | 3000.0 | 2960.2 | 2945.8 | 2942.0 | 2942.4 | 2933.6 |
|  | nce **FAQ**$(D_{\mathrm{cx}})$ | 4737.8 | 4622.8 | 4462.2 | 4056.1 | 3564.9 | 3408.0 | 3352.8 |
| training 0.5 | acc ChFGNN **Proj** | 0.9994 | 0.9962 | 0.9639 | 0.7842 | 0.3400 | 0.1442 | 0.0737 |
|  | acc ChFGNN **FAQ** | 1.0 | 1.0 | 1.0 | 0.9915 | 0.8949 | 0.5105 | 0.1267 |
|  | nce ChFGNN **Proj** | 4736.1 | 4617.1 | 4439.4 | 4025.8 | 3319.0 | 3085.4 | 3038.0 |
|  | nce ChFGNN **FAQ** | 4737.8 | 4629.0 | 4520.0 | 4395.8 | 4188.4 | 3747.2 | 3413.5 |

Table 13: Accuracy **acc** and number of common edges **nce** for Bernoulli graphs as a function of the noise $p_{\mathrm{noise}}$.

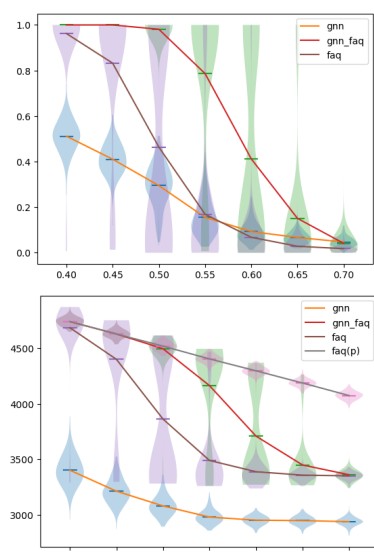

Figure 7: Bernoulli graphs: **acc** (top) and **nce** (bottom) as a function of the noise level. Chained FGNNs were trained at noise level 0.5. **gnn** (resp. **gnn_faq**) for chained FGNNs with **Proj** (resp. **FAQ**) as the last step. **faq** for **FAQ**$(D_{\mathrm{cx}})$ and **faq(p)** for **FAQ**$(\pi^\star)$.

We now explore the generalization properties of the chaining procedure by applying the inference procedure described in Section A.10.2 on datasets with different noise levels. The level of noise used during training (described in Section A.10.1) is 0.5. 7 gives the accuracy **acc** and the number of common edges **nce** for the inference procedure as a function of the noise level. We stop the inference loop when the **nce** obtained after applying **FAQ** to the similarity matrix is not increasing anymore. The red curve gives the performances of our chaining procedure with **FAQ** as the last step, the orange curve gives the performances of our chaining procedure with **Proj** as the last step. We compare our chaining procedure to **FAQ**$(D_{\mathrm{cx}})$ in brown and to **FAQ**$(\pi^\star)$ in grey which corresponds to the maximum number of common edges for these noise levels. The curve for the accuracy of **FAQ**$(D_{\mathrm{cx}})$ is similar to the one obtained in Lyzinski et al. (2015). Our chaining procedure is able to generalize to noise levels different from the one used during training and outperforms **FAQ**$(D_{\mathrm{cx}})$ in all cases. Indeed with a noise level less than 0.5, our chaining procedure recovers the correct permutation for the graph matching problem. Note that we did not try to optimize the performances of our chaining procedure with **Proj** as the last step, and they are indeed increasing if we allow for more loops.

## A.11 Additional results for sparse Erdős-Réyni graphs

Figure 8 gives the performance of our chained GNNs trained at noise level 0.25 for sparse Erdős-Réyni graphs with average degree $d = 4$ and size $n = 500$. We observe that our chaining procedure is able to generalize to noise levels different from the one used during training and outperforms **FAQ**$(D_{\mathrm{cx}})$ in all cases.

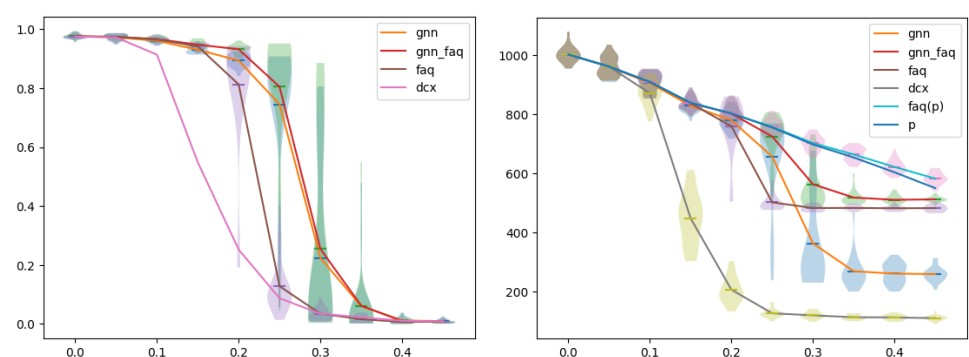

Figure 8: Sparse Erdős-Rényi graphs: **acc** (top) and **nce** (bottom) as a function of the noise level. Chained FGNNs were trained at noise level 0.25. **gnn** (resp. **gnn_faq**) for chained FGNNs with **Proj** (resp. **FAQ**) as the last step. **faq** for **FAQ**$(D_{cx})$ and **faq(p)** for **FAQ**$(\pi^\star)$.

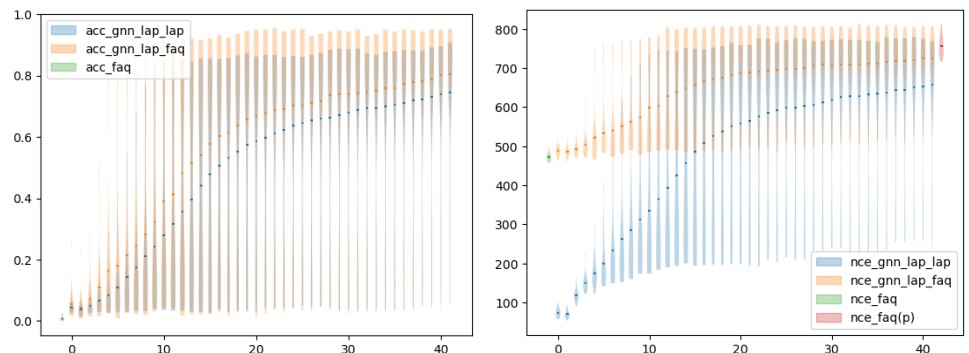

Figure 9: Sparse Erdős-Rényi graphs: **acc** (top) and **nce** (bottom) as a function of the number of iterations $L$ at inference.

Each line in Tables 14 and 15 corresponds to a chained FGNN trained at a given level of noise (given on the left) and tested for all different noises.

Table 14: Accuracy (**acc**) defined in (3) for sparse Erdős-Rényi graphs as a function of the noise $p_{\text{noise}}$. FGNN refers to the architecture in Section A.4 and ChFGNN to our chained FGNNs. **Proj** and **FAQ** are used to produce a permutation (from the similarity matrix computed).

| ER 4 (ACC) | NOISE | 0 | 0.05 | 0.1 | 0.15 | 0.2 | 0.25 | 0.3 | 0.35 |
|---|---|---|---|---|---|---|---|---|---|
| BASELINES | **PROJ**($D_{\text{cx}}$) | 98.0 | 97.3 | 90.3 | 59.3 | 23.3 | 9.1 | 4.0 | 2.0 |
| | **FAQ**($D_{\text{cx}}$) | 98.0 | 97.5 | 96.3 | 94.6 | 72.9 | 13.0 | 3.7 | 1.7 |
| TRAINING 0.05 | CHFGNN **PROJ** | 97.9 | 97.3 | 94.3 | 67.1 | 12.5 | 6.72 | 3.70 | 2.13 |
| | CHFGNN **FAQ** | 97.9 | 97.5 | 96.2 | 72.9 | 43.5 | 10.8 | 4.04 | 1.69 |
| TRAINING 0.10 | CHFGNN **PROJ** | 97.9 | 97.5 | 96.1 | 91.6 | 35.4 | 7.24 | 4.03 | 2.36 |
| | CHFGNN **FAQ** | 97.9 | 97.5 | 96.4 | 94.0 | 38.6 | 12.3 | 4.73 | 1.91 |
| TRAINING 0.15 | CHFGNN **PROJ** | 97.9 | 97.5 | 96.3 | 93.7 | 87.4 | 49.1 | 7.51 | 2.28 |
| | CHFGNN **FAQ** | 97.9 | 97.5 | 96.4 | 94.3 | 90.3 | 54.7 | 9.71 | 1.79 |
| TRAINING 0.20 | CHFGNN **PROJ** | 97.9 | 97.4 | 96.3 | 94.5 | 91.4 | 72.0 | 31.2 | 3.30 |
| | CHFGNN **FAQ** | 97.9 | 97.5 | 96.4 | 95.2 | 93.1 | 76.3 | 35.0 | 3.39 |
| TRAINING 0.22 | CHFGNN **PROJ** | 97.9 | 97.5 | 96.2 | 94.4 | 91.1 | 78.9 | 44.5 | 7.11 |
| | CHFGNN **FAQ** | 97.9 | 97.5 | 96.4 | 95.3 | 93.1 | 82.1 | 48.3 | 7.78 |
| TRAINING 0.24 | CHFGNN **PROJ** | 97.9 | 97.4 | 96.1 | 94.1 | 91.0 | 77.0 | 40.3 | 6.52 |
| | CHFGNN **FAQ** | 97.9 | 97.5 | 96.4 | 95.3 | 93.3 | 80.1 | 43.3 | 6.93 |
| TRAINING 0.26 | CHFGNN **PROJ** | 97.9 | 97.3 | 95.2 | 92.3 | 88.0 | 75.8 | 43.1 | 6.43 |
| | CHFGNN **FAQ** | 97.9 | 97.5 | 96.4 | 95.2 | 93.2 | 82.5 | 48.2 | 6.87 |
| TRAINING 0.28 | CHFGNN **PROJ** | 97.9 | 95.2 | 88.4 | 79.3 | 68.4 | 55.1 | 26.7 | 6.69 |
| | CHFGNN **FAQ** | 97.9 | 97.5 | 96.3 | 94.9 | 92.7 | 82.9 | 38.7 | 7.62 |
| TRAINING 0.30 | CHFGNN **PROJ** | 97.9 | 91.5 | 78.0 | 63.6 | 50.6 | 35.4 | 15.1 | 4.76 |
| | CHFGNN **FAQ** | 97.9 | 97.4 | 96.2 | 94.8 | 92.1 | 73.5 | 23.4 | 5.00 |
| TRAINING 0.35 | CHFGNN **PROJ** | 97.6 | 88.2 | 65.1 | 40.8 | 21.5 | 10.9 | 5.46 | 2.54 |
| | CHFGNN **FAQ** | 97.9 | 97.4 | 96.2 | 94.5 | 68.0 | 19.4 | 5.79 | 2.00 |

## A.12 ADDITIONAL RESULTS FOR DENSE ERDŐS-RÉYNI GRAPHS

For the correlated dense Erdős-Rényi graphs, we used the same dataset as in Yu et al. (2023) with 500 nodes and an average degree of 80. Again, with a noise level of 20%, our chaining GNNs clearly outperform the existing learning algorithms, as we obtain a perfect accuracy (as opposed to an accuracy of zero in Yu et al. (2023) and Chen et al. (2020) without any seed). We see in Table 16 that in this dense setting, **FAQ**($D_{\text{cx}}$) is very competitive but is still slightly outperformed by our chaining FGNNs (orange curve with **Proj** and red curve with **FAQ**, top). In terms of number of common edges, our chained FGNNs does not perform well with **Proj** but performs best with **FAQ**, see Table 16 where the level of noise used for training was 24%.

Figure 10 gives the performance of our chained GNNs trained at noise level 0.24 for sparse Erdős-Rényi graphs with average degree $d = 80$ and size $n = 500$.

Table 15: Number of common edges (**nce**) defined in (4) for sparse Erdős-Rényi graphs as a function of the noise $p_{\text{noise}}$. FGNN refers to the architecture in Section A.4 and ChFGNN to our chained FGNNs. **Proj** and **FAQ** are used to produce a permutation (from the similarity matrix computed).

| ER 4 (NCE) | NOISE | 0 | 0.05 | 0.1 | 0.15 | 0.2 | 0.25 | 0.3 | 0.35 |
|---|---|---|---|---|---|---|---|---|---|
| BASELINES | **PROJ**$(D_{\text{cx}})$ | 997 | 950 | 853 | 499 | 195 | 130 | 115 | 112 |
| | **FAQ**$(D_{\text{cx}})$ | 997 | 950 | 898 | 847 | 723 | 504 | 487 | 485 |
| TRAINING 0.05 | CHFGNN **PROJ** | 997 | 950 | 885 | 630 | 116 | 95 | 87 | 83 |
| | CHFGNN **FAQ** | 997 | 950 | 898 | 761 | 607 | 495 | 485 | 481 |
| TRAINING 0.10 | CHFGNN **PROJ** | 997 | 950 | 897 | 828 | 370 | 99 | 90 | 86 |
| | CHFGNN **FAQ** | 997 | 950 | 899 | 845 | 606 | 501 | 487 | 483 |
| TRAINING 0.15 | CHFGNN **PROJ** | 996 | 950 | 898 | 840 | 768 | 511 | 254 | 86 |
| | CHFGNN **FAQ** | 997 | 950 | 899 | 846 | 791 | 651 | 520 | 484 |
| TRAINING 0.20 | CHFGNN **PROJ** | 996 | 950 | 898 | 846 | 792 | 665 | 456 | 338 |
| | CHFGNN **FAQ** | 997 | 950 | 899 | 849 | 800 | 715 | 596 | 529 |
| TRAINING 0.22 | CHFGNN **PROJ** | 997 | 950 | 898 | 845 | 790 | 694 | 503 | 319 |
| | CHFGNN **FAQ** | 997 | 950 | 899 | 849 | 800 | 730 | 626 | 534 |
| TRAINING 0.24 | CHFGNN **PROJ** | 997 | 950 | 897 | 844 | 789 | 686 | 480 | 296 |
| | CHFGNN **FAQ** | 997 | 950 | 899 | 849 | 800 | 726 | 613 | 527 |
| TRAINING 0.26 | CHFGNN **PROJ** | 997 | 949 | 892 | 834 | 770 | 672 | 499 | 338 |
| | CHFGNN **FAQ** | 997 | 950 | 899 | 849 | 800 | 731 | 626 | 537 |
| TRAINING 0.28 | CHFGNN **PROJ** | 996 | 934 | 836 | 724 | 612 | 504 | 374 | 311 |
| | CHFGNN **FAQ** | 997 | 950 | 899 | 848 | 799 | 732 | 599 | 530 |
| TRAINING 0.30 | CHFGNN **PROJ** | 996 | 897 | 726 | 566 | 446 | 345 | 271 | 246 |
| | CHFGNN **FAQ** | 997 | 950 | 898 | 848 | 797 | 704 | 552 | 513 |
| TRAINING 0.35 | CHFGNN **PROJ** | 995 | 860 | 578 | 347 | 219 | 173 | 159 | 134 |
| | CHFGNN **FAQ** | 997 | 950 | 898 | 847 | 702 | 524 | 494 | 489 |

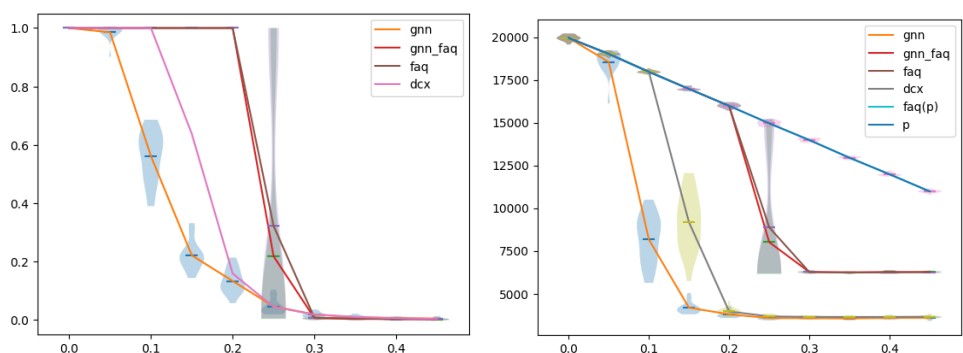

Figure 10: Dense Erdős-Rényi graphs: **acc** (top) and **nce** (bottom) as a function of the noise level. Chained FGNNs were trained at noise level 0.25. **gnn** (resp. **gnn_faq**) for chained FGNNs with **Proj** (resp. **FAQ**) as the last step. **faq** for **FAQ**$(D_{\text{cx}})$ and **faq(p)** for **FAQ**$(\pi^\star)$.

Each line in Tables 16 and 17 corresponds to a chained FGNN trained at a given level of noise (given on the left) and tested for all different noises.

Table 16: Accuracy (**acc**) defined in (3) for dense Erdős-Rényi graphs as a function of the noise $p_{\text{noise}}$. FGNN refers to the architecture in Section A.4 and ChFGNN to our chained FGNNs. **Proj** and **FAQ** are used to produce a permutation (from the similarity matrix computed).

| ER 80 (ACC) | NOISE | 0 | 0.05 | 0.1 | 0.15 | 0.2 | 0.25 | 0.3 | 0.35 |
|---|---|---|---|---|---|---|---|---|---|
| BASELINES | **PROJ**$(D_{\text{cx}})$ | 100. | 100. | 100. | 60.8 | 14.3 | 4.3 | 1.8 | 1.1 |
|  | **FAQ**$(D_{\text{cx}})$ | 100. | 100. | 100. | 100. | 100. | 21.2 | 0.9 | 0.5 |
| TRAINING 0.05 | CHFGNN **PROJ** | 100. | 100. | 99.8 | 16.0 | 5.95 | 2.76 | 1.77 | 1.01 |
|  | CHFGNN **FAQ** | 100. | 100. | 100. | 100. | 54.4 | 1.16 | 0.72 | 0.47 |
| TRAINING 0.10 | CHFGNN **PROJ** | 100. | 100. | 100. | 88.9 | 7.49 | 3.47 | 1.99 | 1.16 |
|  | CHFGNN **FAQ** | 100. | 100. | 100. | 89.0 | 80.0 | 6.58 | 0.85 | 0.52 |
| TRAINING 0.15 | CHFGNN **PROJ** | 100. | 100. | 99.9 | 99.9 | 75.0 | 3.57 | 2.14 | 1.21 |
|  | CHFGNN **FAQ** | 100. | 100. | 100. | 100. | 75.1 | 3.85 | 0.85 | 0.55 |
| TRAINING 0.20 | CHFGNN **PROJ** | 100. | 100. | 100. | 99.9 | 94.0 | 22.9 | 2.06 | 1.22 |
|  | CHFGNN **FAQ** | 100. | 100. | 100. | 100. | 95.0 | 22.7 | 0.83 | 0.52 |
| TRAINING 0.22 | CHFGNN **PROJ** | 100. | 100. | 100. | 99.9 | 97.9 | 49.5 | 2.21 | 1.25 |
|  | CHFGNN **FAQ** | 100. | 100. | 100. | 100. | 99.0 | 50.5 | 1.00 | 0.52 |
| TRAINING 0.24 | CHFGNN **PROJ** | 100. | 99.9 | 93.5 | 83.4 | 67.4 | 34.0 | 2.16 | 1.33 |
|  | CHFGNN **FAQ** | 100. | 100. | 100. | 100. | 98.1 | 57.6 | 0.96 | 0.55 |
| TRAINING 0.26 | CHFGNN **PROJ** | 100. | 99.9 | 78.3 | 39.4 | 13.0 | 3.91 | 2.07 | 1.29 |
|  | CHFGNN **FAQ** | 100. | 100. | 100. | 100. | 94.1 | 5.75 | 0.82 | 0.54 |
| TRAINING 0.28 | CHFGNN **PROJ** | 100. | 99.8 | 70.7 | 31.2 | 9.88 | 3.94 | 2.01 | 1.19 |
|  | CHFGNN **FAQ** | 100. | 100. | 100. | 100. | 84.7 | 12.7 | 0.83 | 0.51 |
| TRAINING 0.30 | CHFGNN **PROJ** | 100. | 99.5 | 62.3 | 24.2 | 8.27 | 3.28 | 1.92 | 1.18 |
|  | CHFGNN **FAQ** | 100. | 100. | 100. | 100. | 80.7 | 3.24 | 0.81 | 0.49 |
| TRAINING 0.35 | CHFGNN **PROJ** | 100. | 96.1 | 47.8 | 18.3 | 6.86 | 3.47 | 1.92 | 1.14 |
|  | CHFGNN **FAQ** | 100. | 100. | 100. | 100. | 69.7 | 8.52 | 0.77 | 0.52 |

## A.13 ADDITIONAL RESULTS FOR REGULAR GRAPHS

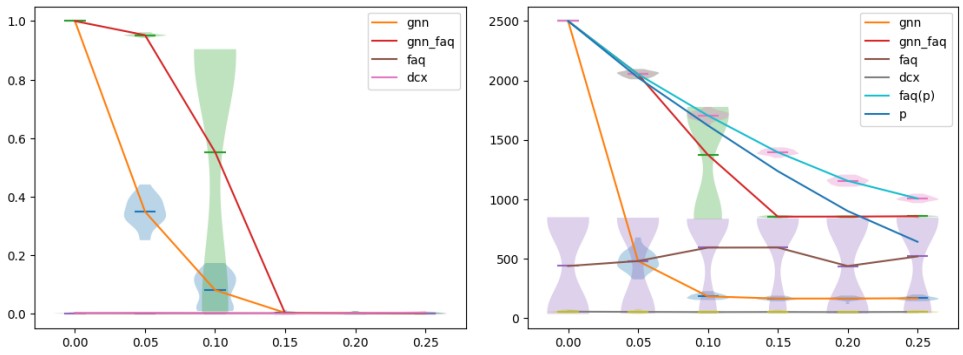

Figure 11: Regular graphs: **acc** (top) and **nce** (bottom) as a function of the noise level. Chained FGNNs were trained at noise level 0.1. **gnn** (resp. **gnn_faq**)for chained FGNNs with **Proj** (resp. **FAQ**) as the last step. **faq** for **FAQ**$(D_{\text{cx}})$, **faq(p)** for **FAQ**$(\pi^{\star})$ and **p** for **nce**$(\pi^{\star})$.

Finally, we propose a new dataset of regular graphs with 500 nodes and an average degree of 10. This is a particularly challenging setting. Indeed, Table 18 shows that **FAQ**$(D_{\text{cx}})$ always fails to solve the graph matching problem here. Similarly, we know that MPNNs are not expressive enough to deal with regular graphs Xu et al. (2018). In view of the following result, we conjecture that using MPNN would not provide a better estimation of the graph matching problem than $D_{\text{cx}}$.

**Theorem A.2.** *Tinhofer (1991) $G_A$ and $G_B$ are fractionally isomorphic, i.e. $\min_{D \in \mathcal{D}_n} \|AD - DB\|_F^2 = 0$, if and only if 1-WL does not distinguish $G_A$ and $G_B$.*

Table 17: Number of common edges (**nce**) defined in (4) for dense Erdős-Rényi graphs as a function of the noise $p_{\text{noise}}$. FGNN refers to the architecture in Section A.4 and ChFGNN to our chained FGNNs. **Proj** and **FAQ** are used to produce a permutation (from the similarity matrix computed).

| ER 80 (NCE) | NOISE | 0 | 0.05 | 0.1 | 0.15 | 0.2 | 0.25 | 0.3 | 0.35 |
|---|---|---|---|---|---|---|---|---|---|
| BASELINES | **PROJ**($D_{\text{cx}}$) | 19964 | 18987 | 17966 | 8700 | 3888 | 3646 | 3633 | 3624 |
| | **FAQ**($D_{\text{cx}}$) | 19964 | 18987 | 17968 | 16990 | 15972 | 7922 | 6272 | 6276 |
| TRAINING 0.05 | CHFGNN **PROJ** | 19964 | 18987 | 17941 | 3794 | 3457 | 3421 | 3429 | 3411 |
| | CHFGNN **FAQ** | 19964 | 18987 | 17968 | 16990 | 11408 | 6244 | 6252 | 6253 |
| TRAINING 0.10 | CHFGNN **PROJ** | 19964 | 18987 | 17968 | 15479 | 3522 | 3459 | 3453 | 3449 |
| | CHFGNN **FAQ** | 19964 | 18987 | 17968 | 15811 | 13935 | 6681 | 6251 | 6257 |
| TRAINING 0.15 | CHFGNN **PROJ** | 19964 | 18987 | 17967 | 16989 | 12842 | 3470 | 3469 | 3456 |
| | CHFGNN **FAQ** | 19964 | 18987 | 17968 | 16990 | 13544 | 6421 | 6254 | 6256 |
| TRAINING 0.20 | CHFGNN **PROJ** | 19964 | 18987 | 17968 | 16990 | 15216 | 6113 | 3483 | 3472 |
| | CHFGNN **FAQ** | 19964 | 18987 | 17968 | 16990 | 15487 | 8172 | 6258 | 6254 |
| TRAINING 0.22 | CHFGNN **PROJ** | 19964 | 18987 | 17968 | 16987 | 15701 | 9189 | 3644 | 3628 |
| | CHFGNN **FAQ** | 19964 | 18987 | 17968 | 16990 | 15876 | 10614 | 6257 | 6263 |
| TRAINING 0.24 | CHFGNN **PROJ** | 19964 | 18969 | 16241 | 13028 | 9561 | 6166 | 3615 | 3591 |
| | CHFGNN **FAQ** | 19964 | 18987 | 17968 | 16990 | 15779 | 11227 | 6258 | 6255 |
| TRAINING 0.26 | CHFGNN **PROJ** | 19964 | 18976 | 12528 | 5795 | 3925 | 3626 | 3591 | 3545 |
| | CHFGNN **FAQ** | 19964 | 18987 | 17968 | 16990 | 15388 | 6515 | 6257 | 6257 |
| TRAINING 0.28 | CHFGNN **PROJ** | 19964 | 18948 | 10846 | 4975 | 3756 | 3587 | 3542 | 3512 |
| | CHFGNN **FAQ** | 19964 | 18987 | 17968 | 16990 | 14424 | 7207 | 6253 | 6256 |
| TRAINING 0.30 | CHFGNN **PROJ** | 19964 | 18861 | 9289 | 4419 | 3651 | 3489 | 3478 | 3472 |
| | CHFGNN **FAQ** | 19964 | 18987 | 17968 | 16990 | 14032 | 6354 | 6254 | 6258 |
| TRAINING 0.35 | CHFGNN **PROJ** | 19964 | 17850 | 6943 | 4003 | 3578 | 3512 | 3492 | 3461 |
| | CHFGNN **FAQ** | 19964 | 18987 | 17968 | 16990 | 12877 | 6853 | 6254 | 6256 |

In contrast, our FGNN architecture defined in Section A.4 is able to deal with regular graphs and our chaining procedure learns the correct information about the graph matching problem when the noise is low enough.

Note that we are in a setting where **FAQ**$(\pi^\star) \neq \pi^*$ as soon as the noise level is above $5\%$ so that $\pi^\star \neq \pi^{A \to B}$. In this case, we believe that $\pi^{A \to B} = $ **FAQ**$(\pi^\star)$ (but to check it we should solve the graph matching problem!).In Figure 11, the training was done with a noise level of $10\%$ so that labels were noisy. Still performances of our chained FGNNs with **FAQ** are very good. We do not know of any other algorithm working in this setting.

Table 18: Accuracy (**acc**) defined in (3) for Regular graphs as a function of the noise $p_{\text{noise}}$. FGNN refers to the architecture in Section A.4 and ChFGNN to our chained FGNNs. **Proj** and **FAQ** are used to produce a permutation (from the similarity matrix computed).

| REGULAR RANDOM GRAPHS WITH DEGREE 10 | | | | | | |
|---|---|---|---|---|---|---|
| REGULAR (ACC) | NOISE | 0 | 0.05 | 0.1 | 0.15 | 0.2 |
| BASELINES | **PROJ**$(D_{\text{cx}})$ | 0.2 | 0.2 | 0.3 | 0.1 | 0.2 |
| | **FAQ**$(D_{\text{cx}})$ | 0.2 | 0.2 | 0.2 | 0.2 | 0.2 |
| TRAINING 0.05 | CHFGNN **PROJ** | 100. | 95.2 | 2.60 | 0.67 | 0.27 |
| | CHFGNN **FAQ** | 100. | 95.6 | 8.31 | 0.49 | 0.24 |
| TRAINING 0.07 | CHFGNN **PROJ** | 100. | 95.3 | 34.6 | 0.70 | 0.27 |
| | CHFGNN **FAQ** | 100. | 95.6 | 36.0 | 0.54 | 0.25 |
| TRAINING 0.09 | CHFGNN **PROJ** | 100. | 95.2 | 54.4 | 0.86 | 0.34 |
| | CHFGNN **FAQ** | 100. | 95.6 | 55.6 | 0.78 | 0.22 |
| TRAINING 0.11 | CHFGNN **PROJ** | 100. | 72.4 | 30.5 | 0.86 | 0.27 |
| | CHFGNN **FAQ** | 100. | 95.6 | 61.8 | 0.70 | 0.25 |
| TRAINING 0.13 | CHFGNN **PROJ** | 79.2 | 16.9 | 2.13 | 0.55 | 0.25 |
| | CHFGNN **FAQ** | 100. | 95.6 | 2.14 | 0.37 | 0.24 |
| TRAINING 0.15 | CHFGNN **PROJ** | 60.4 | 13.3 | 1.69 | 0.52 | 0.30 |
| | CHFGNN **FAQ** | 100. | 95.6 | 1.37 | 0.34 | 0.21 |

Each line in Tables 18 and 19 corresponds to a chained FGNN trained at a given level of noise (given on the left) and tested for all different noises.

Table 19: Number of common edges (**nce**) defined in (4) for Regular graphs as a function of the noise $p_{\text{noise}}$. FGNN refers to the architecture in Section A.4 and ChFGNN to our chained FGNNs. **Proj** and **FAQ** are used to produce a permutation (from the similarity matrix computed).

| REGULAR RANDOM GRAPHS WITH DEGREE 10 | | | | | | |
|---|---|---|---|---|---|---|
| REGULAR (NCE) | NOISE | 0 | 0.05 | 0.1 | 0.15 | 0.2 |
| BASELINES | **PROJ**$(D_{\text{cx}})$ | 51 | 51 | 50 | 49 | 50 |
| | **FAQ**$(D_{\text{cx}})$ | 385 | 425 | 456 | 369 | 496 |
| TRAINING 0.05 | CHFGNN **PROJ** | 2500 | 2034 | 178 | 101 | 100 |
| | CHFGNN **FAQ** | 2500 | 2059 | 901 | 835 | 835 |
| TRAINING 0.07 | CHFGNN **PROJ** | 2500 | 2036 | 741 | 103 | 172 |
| | CHFGNN **FAQ** | 2500 | 2059 | 1193 | 836 | 852 |
| TRAINING 0.09 | CHFGNN **PROJ** | 2500 | 2034 | 1105 | 281 | 95 |
| | CHFGNN **FAQ** | 2500 | 2059 | 1381 | 871 | 836 |
| TRAINING 0.11 | CHFGNN **PROJ** | 2500 | 1343 | 563 | 192 | 114 |
| | CHFGNN **FAQ** | 2500 | 2059 | 1438 | 850 | 837 |
| TRAINING 0.13 | CHFGNN **PROJ** | 1608 | 210 | 108 | 88 | 71 |
| | CHFGNN **FAQ** | 2500 | 2059 | 841 | 836 | 834 |
| TRAINING 0.15 | CHFGNN **PROJ** | 984 | 163 | 96 | 86 | 87 |
| | CHFGNN **FAQ** | 2500 | 2059 | 837 | 836 | 836 |

## A.14 LLM USAGE

Large language models (LLMs) were employed in this work to assist with grammatical and syntactic corrections, to improve the clarity and readability of sentences and paragraphs, and to support the generation of illustrative figures.

## A.15 REPRODUCIBILITY STATEMENT

To ensure reproducibility, we provide the complete codebase used for training and inference, which produces all results reported in this paper. Detailed descriptions of hyperparameters, training procedures, and evaluation metrics are included in the main text and appendix.

