# OpenReview forum: "Bootstrap Learning for Combinatorial Graph Alignment with Sequential GNNs"
_ICLR.cc/2026/Conference — Submitted to ICLR 2026_

### Official Review · Reviewer_JMJ9 · 2025-10-27

**Soundness:** 3
**Presentation:** 3
**Contribution:** 3
**Rating:** 6
**Confidence:** 2

**Summary:**

This paper introduces a novel "chaining" method for the combinatorial Graph Alignment Problem (GAP), which involves finding the optimal node correspondence between two unlabeled graphs using only structural information. The core idea is to train a sequence of Graph Neural Networks (GNNs) where each subsequent GNN learns to refine the similarity matrix produced by its predecessor, leveraging discrete ranking information about node alignment quality. This bootstrap process, combined with a powerful Folklore-type GNN architecture that operates on node pairs, allows the model to iteratively improve its solutions. The method is shown to significantly outperform state-of-the-art optimization-based solvers like FAQ, especially on challenging instances such as regular graphs where traditional methods fail. Furthermore, the chained GNNs can be effectively combined with traditional solvers (e.g., as an initialization for FAQ) to create hybrid models that achieve superior performance.

**Strengths:**

High Novelty and Impact: The "chaining" idea is innovative and represents a significant shift from typical end-to-end learning approaches for CO problems. Successfully outperforming a strong traditional solver like FAQ on its home turf (synthetic benchmarks) is a notable achievement that could influence the field.

Compelling Empirical Results: The experimental evaluation is extensive and rigorous. The results are clear and convincing, showing dramatic improvements over baselines, particularly in high-noise regimes and on regular graphs. The use of synthetic data allows for controlled and unambiguous benchmarking.

Effective Combination of Paradigms: The paper excellently demonstrates how machine learning and traditional optimization can be combined, rather than positioned as rivals. Using the GNN to produce a high-quality initialization for FAQ is a practical and powerful insight.

Strong Ablation Studies: The paper includes valuable analyses on the effect of chain length, the "looping" technique, and the importance of training noise level, providing deep insights into the method's behavior.

Well-Identified Limitations of Prior Work: The introduction and related work sections do a good job of contextualizing the paper by clearly stating the limited success of prior learning-based methods in surpassing traditional solvers for pure CO problems.

**Weaknesses:**

1, Scalability and Computational Cost: The proposed method is computationally intensive. The Folklore-type GNN architecture operates on node pairs, leading to \(O(n^2)\) memory complexity, and the chaining procedure requires sequential training and inference of multiple such networks. While the linear time complexity of inference is mentioned, the practical wall-clock time and memory footprint for large graphs (e.g., n > 10,000) remain a significant concern and are not thoroughly discussed.
2, Limited Theoretical Analysis:  While the method is empirically powerful, it lacks a solid theoretical foundation. The paper does not provide theoretical guarantees on convergence or approximation ratios. Why and how the chaining procedure leads to iterative improvement is explained intuitively but not formally characterized.
3, Evaluation on Real-World Data: The exclusive use of synthetic benchmarks, while methodologically sound for a CO paper, leaves open the question of performance on real-world graphs. Real-world graphs often have features, community structure, and noise patterns that differ from the synthetic models used here. Demonstrating effectiveness on even a few real-world datasets would greatly strengthen the paper's practical claims.
4, Clarity of the "Looping" Mechanism: The "looping" technique (reusing the final GNN \(g^{(L)}\) multiple times) is an interesting empirical finding, but its rationale is somewhat unclear. Why does the final GNN generalize to its own outputs in subsequent iterations? A deeper investigation or discussion of this phenomenon would be beneficial.

**Questions:**

Scalability: What is the largest graph size (n) you can feasibly handle with your current implementation, and what are the primary bottlenecks (memory, training time, inference time)? Are there strategies to make the chaining procedure or the Folklore-GNN more scalable?

Generalization: This method is trained on a specific graph model (e.g., ER) and noise level. How would you expect it to perform on a graph from a completely different distribution (e.g., a small-world or scale-free network)? Have you conducted any out-of-distribution tests?

Theoretical Underpinnings: Can you provide any theoretical intuition or analysis for why the chaining procedure works? For instance, can you frame it as a form of fixed-point iteration or relate it to a known optimization algorithm?

Ablation on Architecture: How critical is the expressive Folklore-GNN architecture to the success of chaining? Could a less expressive but more scalable MPNN be used in the chain if given more steps, or is the high-quality initial output from the Folklore-GNN indispensable for bootstrapping?

---

> ### Author Response · Authors · 2025-11-20
> **Revised version contains new experiments on real graphs confirming our results on synthetic graphs**
>
> We thank the reviewer for the detailed comments.
>
> > 1- Scalability: What is the largest graph size (n) you can feasibly handle with your current implementation, and what are the primary bottlenecks (memory, training time, inference time)? Are there strategies to make the chaining procedure or the Folklore-GNN more scalable?
>
> The primary bottleneck for our chained FGNNs is memory, which scales as
> $O(n^2)$. In our new experiments on real graphs (Section 5.5), the largest graph we could handle has $n=1174$. All experiments were run on a single GPU, and handling larger graphs would be possible with multiple GPUs; however, the runtime for FAQ(D_cx) (used to compare its performances with our chained FGNNs) would then become prohibitive.
> Currently, the inference time of our chained FGNNs (running on GPU) is much faster than FAQ(D_cx) (running on CPU). The bottleneck is not the FAQ algorithm itself—SciPy’s [quadratic_assignment(method='faq')](https://docs.scipy.org/doc/scipy/reference/optimize.qap-faq.html#optimize-qap-faq) provides a highly optimized implementation—but the Frank–Wolfe solver we had to implement to obtain FAQ’s initialization via the convex relaxation. Our unoptimized Python implementation of this step dominates the total runtime.
>
> > 2- Generalization: This method is trained on a specific graph model (e.g., ER) and noise level. How would you expect it to perform on a graph from a completely different distribution (e.g., a small-world or scale-free network)? Have you conducted any out-of-distribution tests?
> > W3 Evaluation on Real-World Data: (...) Demonstrating effectiveness on even a few real-world datasets would greatly strengthen the paper's practical claims.
>
> We ran new experiments on real graphs where noise is added. We observed that our chained FGNNs trained on sparse Erdos Reyni graphs (**ChFGNN ER4** below) have slightly better performances than FAQ. If our chained FGNNs (**ChFGNN** below) are trained on the noisy real graphs then performances are even better. All these results have been added in the new Section 5.5:
>
> - Exp 2a: Number of common edges on noisy versions of **yeast25_Y2H1** (yeast protein-protein interaction network):
>
> nce |  noise 0.05 | noise 0.1 |
> --------|----|-----------|
> **FAQ(D_cx)**|   7660 | 7245
> **ChFGNN ER4** | 7693      |   7297 |
> **ChFGNN**|  7732| 7404|
>
> - Exp 2b: Number of common edges on noisy versions of **ca-netscience** (coauthorship network):
>
> nce | noise 0.1 | noise 0.2 |
> --------|----------|-----------|
> **FAQ(D_cx)**|    822 |  687
> **ChFGNN ER4** |   818     |  688 |
> **ChFGNN**|  824 |  724 |
>
> - Exp 2c: Number of common edges on noisy versions of **inf-euroroad** (road network):
>
> nce | noise 0.1 | noise 0.2 |
> --------|----------|-----------|
> **FAQ(D_cx)**|    1170 |  940
> **ChFGNN ER4** |     1111    |    970  |
> **ChFGNN**|   1213 |  963  |
>
> > 3- Theoretical Underpinnings: Can you provide any theoretical intuition or analysis for why the chaining procedure works? (...)
> > W4 Clarity of the "Looping" Mechanism (...)
>
> We agree that our theoretical understanding of the chaining method remains limited. Conceptually, our approach resembles classical iterative refinement algorithms (such as Frank–Wolfe, which underlies FAQ), but with an important distinction: instead of repeatedly applying a single update rule, we *learn* a sequence of refinement operators $f, g^{(1)}, g^{(2)}, \ldots, g^{(L)}$, each specialized to improve solutions within a specific quality regime. We explored connections to denoising diffusion models in the space of permutations, but did not find a fully satisfying theoretical explanation.
>
> Regarding the looping mechanism, the empirical success of reapplying $g^{(L)}$ suggests that the final operator acts as a form of learned fixed-point iteration: it tends to map its own outputs closer to a stable point, which often corresponds to a high-quality solution of the alignment problem. A deeper theoretical analysis of this behavior is an interesting direction for future work.
>
>
> > 4- Ablation on Architecture: How critical is the expressive Folklore-GNN architecture to the success of chaining? Could a less expressive but more scalable MPNN be used in the chain if given more steps, or is the high-quality initial output from the Folklore-GNN indispensable for bootstrapping?
>
> GNN-based methods for the graph alignment problem are conceptually similar to the first step of our chaining. In the appendix (Section A.1), we include results from recent MPNN-based approaches. These message-passing architectures perform worse than FAQ, whereas only our chained FGNNs are able to surpass FAQ. In particular, regular graphs illustrate a case where MPNNs are inherently limited, making the expressive Folklore-GNN architecture crucial for producing high-quality initial outputs that enable successful chaining.

---

### Official Review · Reviewer_Dh3P · 2025-10-29

**Soundness:** 2
**Presentation:** 3
**Contribution:** 2
**Rating:** 4
**Confidence:** 4

**Summary:**

Combinatorial graph alignment aims to find the optimal permutation that best represents the node correspondence between two graphs. The authors proposed a new iterative learning framework using a sequence of graph neural networks (GNNs). Each GNN refined the solution predicted by the previous GNN by additionally considering the alignment quality of the prior prediction. The process began with an initial permutation prediction, followed by the computation of a ranking score for each node that reflects its alignment quality. This ranking was then incorporated as additional input--alongside the adjacency matrix--to the next GNN, which generates a refined prediction. Furthermore, the authors designed a new GNN architecture, FGNN, which exhibits improved expressivity. Experimental results demonstrated that both FGNN and the proposed iterative learning framework improve alignment accuracy.

**Strengths:**

(S1) The authors introduced a novel learning methodology for combinatorial graph alignment.

(S2) The explanation of the learning framework is clear and easy to follow.

**Weaknesses:**

(W1) Organization of writing:
While the experimental results showed that both the model architecture and the chaining learning mechanism improve alignment accuracy, the architectural details of FGNN are only described in the appendix. This reviewer believes that at least one paragraph describing the model architecture should be included in the main text.

(W2) Lack of runtime comparison:
Runtime is a critical metric for combinatorial optimization problems. Although the authors mentioned that runtime comparison is challenging, they do not provide sufficient justification for this difficulty. According to the appendix, FGNN was inspired by Folklore-type GNNs, which offer higher expressivity at the cost of scalability. Therefore, the use of FGNN may increase computational time, and this aspect should be discussed in more detail.

(W3) Limited baselines:
The authors only compared their approach with non-neural-based optimization methods. To better assess the proposed framework, comparisons with existing GNN-based methods and combinations of GNNs with their iterative learning mechanism should be included. Also, the baseline should include various graph alignment (graph matching) methods in the literature.

(W4) Lack of motivation
The paper did not clearly explain why solving the combinatorial graph alignment problem is important, what broader implications it has, or how plausible the problem setting is where only the adjacency matrix is utilized. A more concrete motivation should be presented in the introduction.

**Questions:**

The authors used ranking-based evaluation to measure alignment quality, which prevents gradient flow. Instead of using ranking, how about introducing a differentiable metric--such as one based on the difference between the softmax of the similarity matrix and the ground-truth permutation matrix?

---

> ### Author Response · Authors · 2025-11-20
> **The revised version addresses the weaknesses raised by the reviewer**
>
> We thank the reviewer for the valuable comments.
>
> > Question: Instead of using ranking, how about introducing a differentiable metric?
>
> We agree that the ranking used in our chaining method is non-differentiable, which forces us to train the FGNNs sequentially. Its purpose, however, is to inject alignment information directly into the node features. While the similarity matrix also encodes alignment information, it represents a mapping from nodes in $G_A$ to nodes in $G_B$, and it is not obvious how to incorporate this cross-graph signal as a differentiable feature within our current Siamese architecture. Designing a differentiable mechanism that preserves this alignment structure is an interesting direction for future work.
>
> > (W1) Organization of writing: (...) at least one paragraph describing the model architecture should be included in the main text.
>
> We agree with the reviewer. In the revised version, we added a short description of the architecture in Section 3.3
>
> > (W2) Lack of runtime comparison
>
> Currently, the inference time of our chained FGNNs (running on GPU) is much faster than FAQ (running on CPU). The bottleneck is not the FAQ algorithm itself—SciPy’s [quadratic_assignment(method='faq')](https://docs.scipy.org/doc/scipy/reference/optimize.qap-faq.html#optimize-qap-faq) provides a highly optimized implementation—but the Frank–Wolfe solver we had to implement to obtain FAQ’s initialization via the convex relaxation. Our Python implementation of this step is not optimized and dominates the total runtime. Because this overhead is specific to our prototype and not inherent to FAQ, we felt that a direct runtime comparison between our GPU-based FGNNs and FAQ with our unoptimized initialization would be misleading, and therefore did not include it.
>
> > (W3) Limited baselines: (...) the baseline should include various graph alignment (graph matching) methods in the literature.
>
> Following the recommendation of Reviewer fXuf, we added new experiments with SGWL (NeurIPS 2019) and FUGAL (NeurIPS 2024) in Section 5.5. These results reinforce our original conclusion: when properly initialized, FAQ outperforms these methods. In our updated experiments, we also reproduced FUGAL’s results on three real-world datasets, and only our chained FGNNs surpass both FAQ and FUGAL.
>
> We further included, in the appendix, comparisons with several GNN-based baselines. As shown in Section A.1, these methods are already outperformed by FAQ.
>
> > (W4) Lack of motivation: (...) A more concrete motivation should be presented in the introduction.
>
> Our primary motivation for studying the graph alignment problem is to explore whether GNN-based methods can improve performance on a purely combinatorial task. The equivariance property of GNNs makes them particularly well-suited for this setting. To illustrate practical relevance, we added experiments on real-world datasets, including protein–protein interaction (PPI) networks in biology, co-authorship networks in social sciences, and road networks in transportation. We added the following sentence in the introduction: "Finally, we confirm the effectiveness and transferability of our method by achieving strong results on three real-world graph pairs (biology, social networks, and road networks), thereby validating our findings from synthetic data."

---

> > ### Comment · Reviewer_Dh3P · 2025-11-27
> > **Further comments**
> >
> > We appreciate your time and efforts in responding to my review. While some of my previous concerns have been addressed, I believe that W2 and W3 still remain insufficiently resolved.
> >
> > For W2, I continue to believe that a runtime comparison is essential, especially given the combinatorial nature of the task. Moreover, since your proposed method is built upon a backbone with higher expressive power, I am concerned that it may be computationally slower than other GNN-based approaches, and this should be clarified or empirically validated.
> >
> > Regarding W3, I also remain unconvinced. SGWL is from 2019, which makes it too outdated to serve as a competitive baseline. Additionally, evaluating against only two baselines is not sufficient given the breadth of related work, and important GNN-based baselines are missing from the comparison.

---

> > > ### Author Response · Authors · 2025-11-27
> > >
> > > Thank you for the continued constructive feedback.
> > >
> > > **Regarding W2:**
> > > We agree that a runtime comparison across architectures would provide valuable insight. Our primary focus in this work was to investigate whether increasing the expressive power of the GNN backbone leads to improved accuracy. For this reason, we prioritized performance evaluation over efficiency analysis. As part of future work, we plan to optimize our architecture and provide a systematic runtime comparison with existing GNN-based and non-GNN baselines.
> > >
> > > **Regarding W3:**
> > > We apologize if our previous response was unclear. In addition to SGWL (NeurIPS 2019) and FUGAL (NeurIPS 2024) (used for comparison on real world datasets in Section 5.5), we included comparisons with SeedGNN (ICML 2023), PGM (VLDB 2015), SGM (Pattern Recognition 2019), and MGCN (KDD 2020). To ensure fairness, we use the exact synthetic datasets from the SeedGNN implementation. Our results (also in Appendix A.1) show that SeedGNN (and other GNN-based baselines like MGCN) performs significantly worse than FAQ and our method:
> > >
> > > acc | sparse ER (noise=0.2) | dense ER (noise=0.2)
> > > ----|----|----
> > > SeedGNN | 0.3% | 0.1%
> > > PGM | 0.2% | 0.1%
> > > SGM | 0.3% | 0.2%
> > > MGCN | 0.1% | 0.1%
> > > FAQ(D_cx) | 73% | 100%
> > > ChFGNN | 93% | 99%
> > >
> > > These experiments confirm our statement in Section 4 that **before our work, FAQ(Dcx) was the state-of-the-art for GAP on correlated random graphs, outperforming all existing learning and GNN approaches.** Since FAQ remains by far the strongest competitor, we keep the main table focused on it and report the additional baselines in the appendix for readability.
> > >
> > > If the reviewer believes other *specific* GNN baselines for GAP are missing, we would appreciate further pointers, as our literature search did not reveal more recent or competitive alternatives.
> > >
> > > References:
> > > - SGWL: Hongteng Xu, Dixin Luo, and Lawrence Carin. Scalable gromov-wasserstein learning for graph
> > > partitioning and matching. Advances in neural information processing systems, 32, 2019.
> > > - FUGAL: Aditya Bommakanti, Harshith R Vonteri, Konstantinos Skitsas, Sayan Ranu, Davide Mottin, and
> > > Panagiotis Karras. Fugal: Feature-fortified unrestricted graph alignment. Advances in Neural
> > > Information Processing Systems, 37:19523–19546, 2024.
> > > - SeedGNN: Liren Yu, Jiaming Xu, and Xiaojun Lin. Seedgnn: graph neural network for supervised seeded graph
> > > matching. In International Conference on Machine Learning, pp. 40390–40411. PMLR, 2023.
> > > - PGM: Ehsan Kazemi, S Hamed Hassani, and Matthias Grossglauser. Growing a graph matching from a
> > > handful of seeds. Proceedings of the VLDB Endowment, 8(10):1010–1021, 2015.
> > > - SGM: Donniell E Fishkind, Sancar Adali, Heather G Patsolic, Lingyao Meng, Digvijay Singh, Vince
> > > Lyzinski, and Carey E Priebe. Seeded graph matching. Pattern recognition, 87:203–215, 2019.
> > > - MGCN: Hongxu Chen, Hongzhi Yin, Xiangguo Sun, Tong Chen, Bogdan Gabrys, and Katarzyna Musial.
> > > Multi-level graph convolutional networks for cross-platform anchor link prediction. In Proceed-
> > > ings of the 26th ACM SIGKDD international conference on knowledge discovery & data mining,
> > > pp. 1503–1511, 2020.

---

### Official Review · Reviewer_fFqf · 2025-10-31

**Soundness:** 4
**Presentation:** 3
**Contribution:** 4
**Rating:** 8
**Confidence:** 4

**Summary:**

In this paper, the authors present a method for the graph alignment problem that uses several chained GNNs. At each step, a similarity matrix is computed, and then a ranking score for the nodes is obtained according to the permutation resulting from this similarity matrix. These rankings are used as features in the following step. The results are promising.

**Strengths:**

The paper is generally well written, and it includes references to several works that employ different methodologies. The idea is simple (and I consider this a strength), and the results are very good.

**Weaknesses:**

I would have liked to see at least one example involving real graphs, as well as a runtime comparison (during inference).

**Questions:**

The accuracy (eq (3)) measures the coincidence between the "optimal" solution $\pi^{A\to B}$ and the permutation $\pi$. However, given two graphs, there may exist multiple permutations that perfectly align them. This is particularly true for certain regular graphs, and, more generally, for graphs that are not “friendly” (see "On convex relaxation of graph isomorphism", PNAS, Aflalo, Bronstein, and Kimmel), or that do not satisfy other more general conditions (see "On spectral properties for graph matching and graph isomorphism problems", Information and Inference: A Journal of the IMA, by Fiori and Sapiro).
In any case, this accuracy should not be used to evaluate the performance of graph alignment methods. I know that asymptotically most graphs under the ER model have trivial automorphism groups and therefore this issue does not arise, but still, I would remove this evaluation metric from the paper, or at least put a comment on that.

In the description of the FAQ method (lines 166-172), it seem like it solves (6), but the solution of (6) is actually a doubly stochastic matrix, and in fact the last step of the FAQ method consists on a projection step with the Hungarian algorithm, just like Proj.

On line 377: for regular graphs, the convex relaxation may return any doubly stochastic matrix in the convex set that contains (among others) the true permutation matrix/matrices and the barycenter. In this case, the convex problem does not have a unique minimizer, and thus the obtained solution depends on the algorithm used.

---

> ### Author Response · Authors · 2025-11-20
> **Real graph experiments have been added to strengthen our results.**
>
> We thank the reviewer for the detailed comments.
>
> > The accuracy (eq (3)) measures the coincidence between the "optimal" solution and the permutation. However, given two graphs, there may exist multiple permutations that perfectly align them(...) In any case, this accuracy should not be used to evaluate the performance of graph alignment methods.
>
> We agree with this comment but we think we need to keep this measure mainly because in some applications, the quantity of interest is indeed the node correspondence as illustrated in our new experiment with real graphs from biology, social networks or road networks. We added the following sentence in the revised version after the definitions of accuracy and number of common edges (nce): "Note that even if the ratio (of nce/nce_opt) is one, the accuracy may still be low if the GAP has no unique solution (as illustrated on real datasets in Section 5.5)."
>
> > In the description of the FAQ method (lines 166-172), it seem like it solves (6), but the solution of (6) is actually a doubly stochastic matrix, and in fact the last step of the FAQ method consists on a projection step with the Hungarian algorithm, just like Proj.
>
> We agree and made it more explicit.
>
> > On line 377: for regular graphs, the convex relaxation may return any doubly stochastic matrix in the convex set that contains (among others) the true permutation matrix/matrices and the barycenter. In this case, the convex problem does not have a unique minimizer, and thus the obtained solution depends on the algorithm used.
>
> We agree. Since the convex-relaxation algorithm starts from the barycenter, it remains stuck at this uninformative solution. We have updated the wording on page 8.
>
> > I would have liked to see at least one example involving real graphs, as well as a runtime comparison (during inference).
>
> We have added several experiments on real graphs from biology, social networks, and road networks in the new Section 5.5. These experiments show that recent methods are still outperformed by FAQ (see Sections A.1 and A.2), and that only our chained FGNNs achieve superior performance.
> Currently, the inference time of our chained FGNNs (running on GPU) is much faster than FAQ (running on CPU). The bottleneck is not the FAQ algorithm itself—SciPy’s [quadratic_assignment(method='faq')](https://docs.scipy.org/doc/scipy/reference/optimize.qap-faq.html#optimize-qap-faq) provides a highly optimized implementation—but the Frank–Wolfe solver we had to implement to obtain FAQ’s initialization via the convex relaxation. Our Python implementation of this step is not optimized and dominates the total runtime. Because this overhead is specific to our prototype and not inherent to FAQ, we felt that a direct runtime comparison between our GPU-based FGNNs and FAQ with our unoptimized initialization would be misleading, and therefore did not include it.

---

### Official Review · Reviewer_fXuf · 2025-11-05

**Soundness:** 1
**Presentation:** 3
**Contribution:** 2
**Rating:** 2
**Confidence:** 5

**Summary:**

This paper proposes a learning-based method for unlabeled graph alignment exploiting a bootstrap chaining effect over sequential GNNs that operate on node pairs. Each GNN refines a similarity matrix produced by its predecessors over iterations. The central claim of the paper is that it is, after all, possible for a GNN-based method to outperform a traditional optimization method on the combinatorial optimization problem of graph alignment. The claim is meant to be corroboreated by a comparison to FAQ, a method for graph alignment introduced in 2015. All work on the graph alignment problem since 2015 is ignored. Experimentation with synthetic data, including a combination with traditional solvers as post-processing, shows an improvement over other GNN-based methods and FAQ.

**Strengths:**

S1. Novel proposal of bootstrapping GNNs.
S2. Novel suggestion of working on node pairs.
S3. Introduction of challenging benchmark of synthetic regular graphs.

**Weaknesses:**

W1. Ignores all work on graph alignment since 2015; recent advances include SGWL [NeurIPS 2019] and FUGAL [NeurIPS 2024], which outperform the FAQ method considered to be state-of-the-art in this paper.
W2. Experiments limited to synthetic data; real-world data pose unique challenges, as recent work has shown.
W3. Post-processing limited to FAQ and a straightforwards projection by the Hungarian algorithm.

**Questions:**

How does the proposal perform compare to state of the art methods?

---

> ### Author Response · Authors · 2025-11-20
> **New experiments on real graphs show that our proposal performs better than state-of-the-art methods**
>
> We thank the reviewer for the valuable comments.
>
> We agree that we missed the references you pointed out and we added them in the revision:
> - [1] Scalable Gromov-Wasserstein learning for graph partitioning and matching by H Xu, D Luo, L Carin [NeurIPS 2019]
> - [2] FUGAL: Feature-fortified Unrestricted Graph Alignment by Aditya Bommakanti, Harshith Reddy Vonteri, Konstantinos Skitsas, Sayan Ranu, Davide Mottin, Panagiotis Karras [NeurIPS 2024]
>
> Since FUGAL gets better results than SGWL, we used the code from the authors of FUGAL available at: https://github.com/idea-iitd/Fugal ro run FUGAL on our synthetic datasets. Despite trying several possible hyperparameters, we were surprised to get worse results with FUGAL than with FAQ as shown below for the number of common edges:
> |nce |  FUGAL | FAQ | ChFGNN |
> |-----|----|----------|----------|
> |ER sparse (noise 0.1)| 560 | 898| 899|
> |ER sparse (noise 0.2) | 396 | 723 |800 |
> |ER dense (noise 0.1)| 17968 | 17968 | 17968 |
> |ER dense (noise 0.2) | 5853 |15972 | 15876 |
> |Regular (noise 0.05)| 131 | 425 | 2059 |
>
> We then ran the following experiments on the real-world datasets used in [2].
>
> - Exp 1: yeast protein-protein interaction (PPI) network and its noisy variants. It contains a base graph composed of 1,004 nodes (proteins) and 8,323 edges (PPIs), where each edge denotes a trusted interaction and 5 noisy graph variants, each generated by injecting an extra $q\%$ of low-confidence edges into the base network, with $q$ taking values in $\{5, 10, 15, 20, 25\}$. We found that the numbers reported in [2] for FAQ are likely invalid, most probably due to an improper initialization. When FAQ is initialized with the solution of the convex relaxation of GAP, we obtain significantly better results than those in [2]; in fact, FAQ consistently outperforms FUGAL on these datasets. Results are shown below where ChFGNN ER4 corresponds to our chained FGNNs trained on sparse Erdos-Eyni graphs and ChFGNN corresponds to our chained FGNNs trained on the pair obtained from the first datasets. We are giving both the accuracy and the number of common edges.
>
> |      acc/nce           |  yeast5_Y2H1 | yeast10_Y2H1 | yeast15_Y2H1 | yeast20_Y2H1 | yeast25_Y2H1 |
> |------------------------|-------------|--------------|--------------|--------------|--------------|
> | **FAQ Base**    | 37.5 / 7383   | 34.4 / 7245   | 29.1 / 6807   | 23.9 / 6689   | 36.4 / 7383   |
> | **SGWL** [1]   | 83.6 / -- | -- / -- | 66.6 /-- | -- / -- | 58.8/ -- |
> | **FUGAL** [2]    | 83.0 / 8311   | 77.7 / 8231   | 74.3 / 8172   | 70.9 / 8148   | 68.6 / 8095   |
> | **FAQ(D_cx)**    | 84.2 / 8323   | 82.6 / 8317   | 78.0 / 8289   | 77.0 / 8294   | 76.1 / 8306   |
> | **ChFGNN ER4**  | 80.3 / 8300 | 75.3 / 8288 | 67.2 / 8252 | 63.1 / 8213 | 53.1 / 8080 |
> | **ChFGNN**     | training | training | training | 72.2 / 8300  |  69.8 / 8291 |
>
> - Exp 2: To obtain more challenging benchmarks, we also applied the edge-addition–removal noise model to the yeast PPI network with $q=25\%$, the **ca-netscience** coauthorship network, and the **inf-euroroad** road network used similarly in [2]. We obtained results for FUGAL in line with those in [2] but as above, FAQ (when initialized properly) gets much better results than those reported in [2].
>
> - Exp 2a: Performances on noisy versions of **yeast25_Y2H1**:
>
> acc/nce |  noise 0.05 | noise 0.1 |
> --------|----|-----------|
> **FUGAL** | 53.1 / 7480 | 44.6 / 7035|
> **FAQ(D_cx)**|  49.8 / 7660 | 44.7 / 7245
> **ChFGNN ER4** | 47.6 / 7693      |  42.3 / 7297 |
> **ChFGNN**|  54.1 / 7732| 51.3 / 7404|
>
> - Exp 2b: Performances on noisy versions of **ca-netscience**:
>
> acc/nce | noise 0.1 | noise 0.2 |
> --------|----------|-----------|
> **FUGAL** | 60.3 / 794 | 37.7 / 629 |
> **FAQ(D_cx)**|   65.2 / 822 | 45.6 / 687
> **ChFGNN ER4** |   63.5 / 818     |  44.1 / 688 |
> **ChFGNN**|  65.4 / 824 | 57.0 / 724 |
>
> - Exp 2c: Performances on noisy versions of **inf-euroroad**:
>
> acc/nce | noise 0.1 | noise 0.2 |
> --------|----------|-----------|
> **FUGAL** | 18.3 / 818 | 2.9 / 714 |
> **FAQ(D_cx)**|   55.8 / 1170 | 10.9 / 940
> **ChFGNN ER4** |    40.0 / 1111    |   7.5 / 970  |
> **ChFGNN**|  63.5 / 1213 |  15.4 / 963  |
>
> As shown by all our experiments above, only our chained FGNNs are able to get better results than FAQ(D_cx) which outperforms FUGAL. We added these new results on real graphs in the revised version of the paper (Section 5.5) with more details about the comparison with FUGAL in the appendix (Section A.2).
>
> > W3. Post-processing limited to FAQ and a straightforwards projection by the Hungarian algorithm.
>
> We view the compatibility and complementarity of our chained FGNNs with existing CO algorithms as a strength rather than a limitation.

---

> > ### Comment · Reviewer_fXuf · 2025-11-28
> > **Initialization gives FAQ an unfair advantage**
> >
> > The main rebuttal argument seems to be that FAQ ought to be initialized differently than it has been in prior work.
> > This argument misfires in two ways:
> >
> > 1. The provided code appears to initialize FAQ in the supposedly improper way ("with uniform doubly stochastic matrix").
> >
> > 2. The proposed initialization "with the solution of the convex relaxation of GAP" appears to offer FAQ an exclusive advantage that is not granted to other methods, making the comparison asymmetric.

---

> > > ### Author Response · Authors · 2025-11-28
> > >
> > > Thank you for the comments. We address each point below.
> > >
> > > > P1- The main rebuttal argument seems to be that FAQ ought to be initialized differently than it has been in prior work.
> > >
> > > Our rebuttal argument is not that prior work ought to have initialized FAQ differently.
> > > Our argument is simply that:
> > > - FAQ, when initialized according to the procedure recommended in the original theoretical work, is state-of-the-art on the datasets we study,
> > > - and therefore FAQ is the appropriate baseline for our method, which is why the main paper focuses on that comparison.
> > >
> > > We further provide comparisons with more recent (but empirically weaker) methods in the Appendix for completeness.
> > >
> > > We are not asserting that earlier papers should have initialized FAQ differently. Our point is simply that, when FAQ is run using the initialization recommended in its original theoretical analysis, it empirically achieves state-of-the-art performance on our datasets.
> > >
> > > > P2- The provided code appears to initialize FAQ in the supposedly improper way ("with uniform doubly stochastic matrix").
> > >
> > > This is not the case.
> > >
> > > The initialization we use is exactly the one advocated in the original theoretical analysis of FAQ:
> > > - Vince Lyzinski et al., Graph matching: Relax at your own risk, IEEE TPAMI 2015.
> > >
> > > That work states:
> > > > "These theoretical results suggest that initializing the indefinite algorithm (i.e. FAQ) with the convex optimum (i.e. D_cx) might yield improved practical performance. Indeed, experimental results illuminate and corroborate these theoretical findings (...)"
> > >
> > > SciPy’s `quadratic_assignment(method='faq')` does not expose this initialization option and instead uses a uniform doubly-stochastic matrix by default, which the TPAMI paper explicitly calls suboptimal.
> > > Because of this limitation, we compute the convex relaxation optimum ourselves using a Frank–Wolfe routine
> > > `relaxed_normAPPB_FW`, then pass it to FAQ as `P0`:
> > > ```python
> > > # (line 100)
> > > P, col, s = relaxed_normAPPB_FW(A0, A1, max_iter=max_iter, verbose=True)
> > >
> > > # (lines 108-110)
> > > res_qap = quadratic_assignment(
> > >         A1, -A0, method="faq", options={"P0": P, "maxiter": 100}
> > >     )
> > > ```
> > > This directly mirrors the MATLAB implementation provided by the FAQ authors themselves: https://github.com/jovo/FastApproximateQAP/blob/master/code/SGM/relaxed_normAPPB_FW_seeds.m
> > >
> > > Thus, the code in our submission does not use uniform initialization—it uses the theoretically supported initialization recommended by the FAQ authors.
> > >
> > > > P3- The proposed initialization "with the solution of the convex relaxation of GAP" appears to offer FAQ an exclusive advantage that is not granted to other methods, making the comparison asymmetric.
> > >
> > > The initialization is not an “extra advantage”; it is an integral part of the FAQ algorithm as defined by its authors.
> > > FAQ consists of two stages:
> > > - Solve the convex relaxation (via Frank–Wolfe method).
> > > - Use this solution as the initialization for the indefinite refinement step.
> > >
> > > This is not a modification or enhancement we introduce; it is the algorithm described by Vogelstein et al. (PLOS 2015) and analyzed in Lyzinski et al. (TPAMI 2015).
> > >
> > > Importantly:
> > > - The convex relaxation uses only the two input adjacency matrices and does not incorporate any additional information, labels, seeds, or supervision.
> > > - Following this procedure does not provide FAQ with any information that other baselines do not have access to.
> > > - It simply ensures that FAQ is evaluated in the form recommended by its authors and supported by theory.
> > >
> > > In contrast, SciPy’s default implementation is a simplified version that omits this first stage, which explains the weaker performance frequently reported in recent works.
> > >
> > > Our comparison is therefore not asymmetric: we evaluate FAQ as originally intended and documented by its authors.

---

### Author Response · Authors · 2025-11-27
**Python code for FAQ (1)**

Two reviewers requested additional comparisons with algorithms other than FAQ. We appreciate this concern, but we believe FAQ remains the state-of-the-art method for the graph alignment problem. As stated in the paper, several recent works on graph alignment do not, in our view, provide a fair comparison to FAQ (Vogelstein et al., 2015), and FAQ’s performance has often been underestimated due to suboptimal initialization strategies. Our experiments demonstrate that FAQ is still an exceptionally strong baseline, and that our method is, to the best of our knowledge, the first since 2015 to achieve a substantial improvement over it. Because this is a strong and potentially sensitive claim, we welcome careful scrutiny.

Below, we provide the Python code needed to reproduce the FAQ results reported in the last column of Table 11 in our paperfor the Yeast PPI networks. We hope this will help substantiate our claims and strengthen confidence in our contributions.

Here is the result obtained on a standard laptop after downloading the 2 graphs (but you can run the code in Colab instead)
```
Loading adjacency matrices...
Graph sizes: A0=(1004, 1004), A1=(1004, 1004)

Phase 1: Running relaxed_normAPPB_FW...
  Iterations: 500/500
  Execution time: 130.6227 seconds

Phase 2: Running quadratic_assignment (FAQ)...
  Execution time: 0.3720 seconds
  Total time: 130.9947 seconds

Results:
  NCE (Number of Common Edges): 8306
  Accuracy: 0.7610
  Note: FUGAL [NeurIPS 2024] reports FAQ accuracy ~0.5 (see Fig 1 MultiMagna)
```

---

> ### Author Response · Authors · 2025-11-27
> **Python code for FAQ (2)**
>
> Here is the python code:
> ```python
> # in a notebook uncomment the following lines to download the data files
> # !wget -O yeast0_Y2H1.txt https://raw.githubusercontent.com/idea-iitd/Fugal/main/data/real%20noise/MultiMagna/yeast0_Y2H1.txt
> # !wget -O yeast25_Y2H1.txt https://raw.githubusercontent.com/idea-iitd/Fugal/main/data/real%20noise/MultiMagna/yeast25_Y2H1.txt
>
> import time
> import numpy as np
> import pandas as pd
> from scipy.optimize import linear_sum_assignment, quadratic_assignment
>
>
> def get_adj(file):
>     df = pd.read_csv(file, sep=" ", header=None, names=["source", "target"])
>     nodes = sorted(set(df["source"]) | set(df["target"]))
>     node_to_idx = {node: idx for idx, node in enumerate(nodes)}
>     n = len(nodes)
>     adj_matrix = np.zeros((n, n), dtype=int)
>     sources = df["source"].map(node_to_idx).values
>     targets = df["target"].map(node_to_idx).values
>     adj_matrix[sources, targets] = 1
>     adj_matrix[targets, sources] = 1
>     return adj_matrix
>
>
> def perm2mat(p):
>     n = len(p)
>     P = np.zeros((n, n))
>     P[np.arange(n), p] = 1
>     return P
>
>
> def fro_norm(P, A, B):
>     diff = A @ P - P @ B
>     return np.linalg.norm(diff, ord="fro") ** 2
>
>     return -np.trace((A @ P).T @ (P @ B))
>
>
> def relaxed_normAPPB_FW(A, B, max_iter=1000, tol=5e-2, tol_var=1e-4, verbose=False):
>     """Frank-Wolfe algorithm for graph matching via ||AP - PB||_F minimization.
>
>     Args:
>         A, B: Adjacency matrices to match
>         max_iter: Maximum number of iterations
>         tol: Tolerance for objective value
>         tol_var: Tolerance for objective variance between iterations
>         verbose: If True, return number of iterations
>
>     Returns:
>         P: Relaxed permutation matrix (transposed)
>         col_ind: Discrete permutation from rounding P
>         s: Number of iterations (if verbose=True, else None)
>     """
>     # Precompute constant matrices for efficiency
>     AtA = A.T @ A
>     BBt = B @ B.T
>     n = A.shape[0]
>     # Initialize with uniform doubly stochastic matrix
>     P = np.ones((n, n)) / n
>     # Compute initial objective
>     f = fro_norm(P, A, B)
>     var = 1
>     s = 0
>     while f >= tol and var > tol_var and s < max_iter:
>         fold = f
>         # Compute gradient: 2(A^T A P - A^T P B - A P B^T + P B B^T)
>         grad = 2 * (AtA @ P - A.T @ P @ B - A @ P @ B.T + P @ BBt)
>         # Solve linear assignment to find descent direction
>         _, col_ind = linear_sum_assignment(grad)
>         Ps = perm2mat(col_ind)
>         # Line search: find optimal step size
>         C = A @ (P - Ps) + (Ps - P) @ B
>         D = A @ Ps - Ps @ B
>         aq = np.trace(C @ C.T)
>         bq = np.trace(C @ D.T + D @ C.T)
>         # Optimal step size (clamped to [0, 1])
>         alpha = np.clip(-bq / (2 * aq), 0, 1)
>         # Update P with convex combination
>         P = alpha * P + (1 - alpha) * Ps
>         # Update convergence metrics
>         f = fro_norm(P, A, B)
>         var = abs(f - fold)
>         s += 1
>     # Round to discrete permutation
>     _, col_ind = linear_sum_assignment(-P.T)
>     return (P.T, col_ind, s) if verbose else (P.T, col_ind, None)
>
>
> def main():
>     # Load adjacency matrices
>     print("Loading adjacency matrices...")
>     A0 = get_adj("yeast0_Y2H1.txt")
>     A1 = get_adj("yeast25_Y2H1.txt")
>     print(f"Graph sizes: A0={A0.shape}, A1={A1.shape}")
>
>     max_iter = 500
>
>     # Phase 1: Frank-Wolfe relaxation
>     print("\nPhase 1: Running relaxed_normAPPB_FW...")
>     start_time = time.time()
>     P, col, s = relaxed_normAPPB_FW(A0, A1, max_iter=max_iter, verbose=True)
>     relaxed_time = time.time() - start_time
>     print(f"  Iterations: {s}/{max_iter}")
>     print(f"  Execution time: {relaxed_time:.4f} seconds")
>
>     # Phase 2: FAQ refinement
>     print("\nPhase 2: Running quadratic_assignment (FAQ)...")
>     start_time = time.time()
>     res_qap = quadratic_assignment(
>         A1, -A0, method="faq", options={"P0": P, "maxiter": 100}
>     )
>     qap_time = time.time() - start_time
>     print(f"  Execution time: {qap_time:.4f} seconds")
>     print(f"  Total time: {relaxed_time + qap_time:.4f} seconds")
>
>     # Evaluate results
>     print("\nResults:")
>     pred = res_qap["col_ind"]
>
>     # Number of Common Edges
>     nce = (A1 * A0[pred, :][:, pred]).sum() / 2
>     print(f"  NCE (Number of Common Edges): {nce:.0f}")
>
>     # Accuracy (assumes ground truth is identity mapping)
>     acc = np.mean([i == m1 for i, m1 in enumerate(pred)])
>     print(f"  Accuracy: {acc:.4f}")
>     print(
>         f"  Note: FUGAL [NeurIPS 2024] reports FAQ accuracy ~0.5 (see Fig 1 MultiMagna)"
>     )
>
>     return res_qap, P, nce, acc
>
>
> if __name__ == "__main__":
>     main()
> ```

---

### Author Response · Authors · 2025-11-28
**Added section on time-performance trade-off**

We added Section A.2 which analyzes the trade-off between computation time and performance for several iterative graph-matching algorithms, including Frank–Wolfe–based relaxations ($D_{\mathrm{cx}}$ and FAQ), FUGAL, and our chained FGNN models. Because these methods are iterative, their runtime can be controlled by adjusting the number of iterations. Using sparse Erdős–Rényi graphs with noise level of 0.2, we compare their Pareto curves, measuring performance via the number of common edges and recording runtime over 100 graph pairs of size n=500. FUGAL and FAQ are executed on CPU, while chained FGNNs run on GPU. The results show that chained FGNNs achieve superior performance at lower computation time compared to both FAQ and FUGAL.

We hope this new section addresses the last remaining concerns of Reviewers fFqf and Dh3P.

---

### Author Response · Authors · 2025-12-03
**Final note to the Area Chair**

We would like to sincerely thank all four reviewers and the previous Area Chair for their careful reading of our submission and for the many constructive comments. We have prepared a revised version of the paper in which all modifications are highlighted in red in both the main text and the appendix. Below we briefly summarize the main additions made to address all points raised during the review process:

1. **Experiments on real graphs (Section 5.5).**
We now include evaluations on three categories of real-world networks: biology, social sciences, and transportation. These experiments demonstrate that the empirical findings reported in the original submission also hold on real data.

2. **Comparison with additional competitors (Appendix A.1 and A.3).**
    - Appendix A.1: We add comparisons with SeedGNN (ICML 2023), PGM (VLDB 2015), SGM (Pattern Recognition 2019), and MGCN (KDD 2020), using the exact synthetic datasets released with the SeedGNN implementation to ensure fairness.
    - Appendix A.3: We add comparisons with S-GWL (NeurIPS 2019) and FUGAL (NeurIPS 2024), using the real datasets employed in those papers so our results can be directly compared to the authors’ reported numbers.

3. **Time–performance trade-off (Appendix A.2).**
We now provide an empirical analysis of the computational cost versus accuracy for FAQ, FUGAL, and our proposed chained FGNNs.

Our work challenges a trend in recent graph alignment literature by showing that several recent methods do not compare fairly against the FAQ algorithm (Vogelstein et al., 2015) and that FAQ’s performance has often been underestimated due to suboptimal initialization strategies. Across all our experiments—including the new evaluations added in this revision—properly initialized FAQ remains an exceptionally strong baseline. Moreover, our chained FGNNs consistently outperform FAQ while requiring less computation time.

We believe the revised submission significantly strengthens our initial claims and clarifies the methodological contributions of our work.

---

### Meta-Review · Area_Chair_XsPv · 2026-01-04

**Summary:**

The paper proposes a sequential GNN framework for graph alignment with a bootstrap learning effect. While the idea is novel, several concerns remain unresolved:

* **Runtime and Scalability:** The combinatorial nature of graph alignment requires clear runtime comparisons. The added appendix plot is insufficient, and efficiency relative to other GNN baselines is not convincingly demonstrated.

* **Initialization Fairness:** The claim that FAQ requires convex‑optimum initialization raises questions of fairness, since similar strategies could be applied to other QAP approximations. The comparison risks being asymmetric.

* **Baselines:** Important baselines were initially missing and later primarily added only in appendices, without systematic integration into the main experimental narrative. This weakens the empirical evidence.

Overall, the contribution is interesting but does not yet meet the bar for ICLR due to lack of rigorous runtime analysis, fairness in baseline treatment, and integration of comparisons.

**Reviewer Concerns:**

While the paper introduces an interesting chaining procedure of sequential GNNs for graph alignment, several critical issues remain unresolved after rebuttal and revision:

**Runtime Comparison**: The combinatorial nature of graph alignment makes runtime and scalability essential evaluation criteria. Although the authors added a time–performance trade‑off plot in the appendix, the integration is limited and does not convincingly address concerns about computational efficiency. Given that the proposed method leverages a more expressive backbone, it is plausible that it may be slower than other GNN‑based approaches. This requires systematic empirical validation, which is currently missing. Despite being asked during the rebuttal, the authors chose not to provide this comparison.

**Initialization Strategy for FAQ**: The authors argue that FAQ’s performance has been underestimated due to suboptimal initialization and advocate for convex‑optimum initialization. However, this raises the question of why similar initialization strategies cannot be applied to other QAP approximations such as FUGAL. Without a fair and symmetric treatment of baselines, the comparison risks being biased in favor of FAQ.

**Missing and Poorly Integrated Baselines**: Several important baselines (e.g., SGWL, FUGAL) were initially omitted. While some were added during rebuttal, they are not systematically integrated into the main experimental narrative. This fragmented presentation weakens the strength of the empirical evidence and makes it difficult to assess the true comparative performance of the proposed method.

Some of the concerns there addressed are as follows:

* It is an indeed an interesting observation that FAQ improves dramatically with the special initialization. Much of the baselines missed this initialization since (1) it was proposed in a different work, and it is not strongly advocated for in the official codebase. So this work gives prominence to this important optimization aspect.

* The revised submission included evaluations on real‑world graphs, showing that the proposed method’s improvements also hold beyond synthetic benchmarks.

**Reviewer Scores:**

Two reviewers were positive and are likely to maintain their original scores. The other two remain negative. The reviewer who initially gave a rating of 2 may at best raise it to 4, but is unlikely to turn positive. Their main concern was why the initialization strategy that benefits FAQ could not also be applied to other QAP methods such as FUGAL, and the authors’ follow‑up did not adequately address this. The reviewer with a rating of 4 requested runtime comparisons, but the authors did not integrate these results into the main paper.

---

### Decision · Program_Chairs · 2026-01-26

Reject